# Encoding of visual stimuli and behavioral goals in distinct anatomical areas of monkey ventrolateral prefrontal cortex

Claudio Basile[1☉], Marzio Gerbella[1☉], Alfonso Gravante[2], Amelia Lapadula[1], Francesca Rodà[3], Luciano Simone[1], Leonardo Fogassi[1]*, Stefano Rozzi[1]*

1 Department of Medicine and Surgery, University of Parma, Parma, Italy, 2 Section on Cognitive Neurophysiology and Imaging, National Institute of Mental Health, Bethesda, Maryland, United States of America, 3 Clinical and Experimental Medicine PhD Program, University of Modena and Reggio Emilia, Modena, Italy

☉ These authors contributed equally to this work.
* leonardo.fogassi@unipr.it (LF); stefano.rozzi@unipr.it (SR)

## Abstract

The lateral prefrontal cortex has been classically defined as an associative region involved in the so-called executive functions, such as guiding behavior based on abstract rules and mnemonic information. However, most neurophysiological studies on monkeys did not address the issue of whether distinct anatomical sectors of lateral prefrontal cortex play different functional roles. The main aim of this work is to study functional properties of neurons recorded from a large part of ventrolateral prefrontal cortex (VLPF) of two monkeys performing passive visual tasks and a visuo-motor task, and to map them on the anatomical areas defined on the basis of our recent parcellations. Our results show that some functional features are broadly distributed within VLPF, while others characterize specific areas. In particular, the temporal structuring of events and the general behavioral rule appear to be coded in all recorded areas, while each area differently contributes to the encoding of visual features and to the exploitation of contextual information for guiding behavior. Caudal VLPF areas, and especially caudal 12r, are characterized by a strong coding of visual information, both when passively presented or exploited for guiding behavior, while middle VLPF areas, and especially middle 46v, are rather more involved in the processing of contextual information for action organization. In this latter sector, visual stimuli/instructions appear to be encoded in a pragmatic format, that is, in terms of the associated behavioral outcome. Finally, area 45A and more anterior VLPF areas are characterized by a generally lower responsiveness to the employed tasks. Altogether, our findings indicate that caudal VLPF areas represent the first processing stage of visual input while middle VLPF areas primarily contribute to the selection and planning of contextually appropriate behaviors.

**Data availability statement:** All Data files and codes are available from the OFS database (accession link: https://osf.io/j8fcs/), in the folders DATA and CODE, respectively.

**Funding:** This work was supported by the Italian Ministry of University and Research (MUR) with the following grants: [PRIN 2015, code: 2015AWSW2Y_005] https://prin.mur.gov.it/ to L.F., [PRIN 2020, Code: 20208RB4N9] https://prin.mur.gov.it/ to L.F., and #NEXTGENERATIONEU (NGEU), National Recovery and Resilience Plan (NRRP), project MNESYS (PE0000006)—A Multiscale integrated approach to the study of the nervous system in health and disease (DN.1553 11.10.2022) https://mnesys.eu/ to L.F. The funders had no role in study design, data collection and analysis, decision to publish, or preparation of the manuscript.

**Competing interests:** The authors have declared that no competing interests exist.

**Abbreviations:** BM, biological movement; HG, human grasping; HM, human mimicking; LPF, lateral prefrontal cortex; OM, object motion; VLPF, ventrolateral prefrontal cortex.

## Introduction

The ability to adapt goal-directed behavior to a continuously changing context is fundamental for complex social animals such as human and non-human primates. Lateral prefrontal cortex (LPF) plays a key role in this function, controlling a series of processes collectively defined as "executive functions", which allow to select, organize, and optimize behaviors in order to reach specific intended goals [1–3]. Most classical studies on executive functions in monkeys have examined LPF as a whole. They found that neurons involved in high-order functions, such as attention, working memory, rule coding, goal coding, and decision-making, are broadly distributed in this region [2,4–10]. However, the functional complexity [11–17] and anatomical inhomogeneity [18–25] of the LPF suggest that it should not be viewed as a single anatomical and functional structure, but rather as comprising relatively distinct sectors. In particular, various studies demonstrated an anatomo-functional distinction between the dorsal and ventral sectors of the LPF, and along the rostro-caudal axis [14–16,23,24,26–28].

Dorso-ventral functional differences have been mainly demonstrated by electrophysiological studies focused on working memory processes and tasks requiring saccadic movements as output responses, showing a stronger involvement of the dorsal sector in processing spatial information and of the ventral one in processing feature-related information [29–32]. Rostro-caudal differences in the distribution of functional properties can be inferred from studies employing different type of motor behaviors, which showed that oculomotor responses are typically found in more caudal regions, while responses related to forelimb movements are usually found more rostrally within LPF [33–37]. Constantinidis and coworkers [15] also described rostro-caudal functional differences in visual processing, with posterior areas more involved in coding visual information per se, and more anterior regions having a stronger role in exploiting it to guide behavior.

Anatomical evidence in the monkey shows a complex organization of LPF, with patterns of connections that, although partly overlapping, change along dorso-ventral and rostro-caudal axes [19,22,23,25,38–41]. Concerning the ventrolateral prefrontal cortex (VLPF), the connections with the temporal lobe increase ventrally, suggesting a strong role of the ventralmost territory in processing visual information, while those with the parietal and premotor areas increase dorsally, suggesting a major role of this region in controlling motor behavior. On the other hand, VLPF can also be subdivided into at least three vertical strips: a caudal one, strongly connected with parietal and frontal areas and subcortical centers involved in oculomotor control, an intermediate strip, strongly connected with cortical and subcortical structures involved in the control of reaching and grasping and a rostral strip, mostly characterized by intrinsic prefrontal connections. To the best of our knowledge, only the pioneering study of Tanila and coworkers mapped the distribution of specific functional properties and correlated this distribution with connectional data, suggesting the presence of some type of functional specialization within LPF both on the dorso-ventral and on the rostro-caudal axis [34].

Altogether, these data indicate that some functional and anatomical features are broadly distributed along VLPF, while others appear to be localized in specific sub-regions/areas. This is in line with theoretical models focused on describing the adaptability of functional properties in the prefrontal cortex (e.g., adaptive coding and multiple demand framework models, [42–44]). In particular, Duncan and coworkers observe that, although neurons encoding different types of information (e.g., location and object) are broadly distributed across LPF, different subregions are characterized by a maximal sensitivity to one specific type of information. They propose that the use of complex tasks allows one to identify the broader distribution of properties, while low-demanding behavioral paradigms could highlight regional specialization.

The general aim of this work is to map the distribution of VLPF neuronal properties based on the hypothesis that functional specialization depends on the specific connections that characterize each anatomical area. To achieve this aim: (1) we recorded single neuron activity in monkeys performing different tasks investigating how visual information is either passively processed or exploited to guide a behavior involving the decision to produce or withhold grasping actions; (2) we analyzed the recorded data with reference to the anatomical parcellations produced by our group [20–22,40].

## Results

In the results section we will (1) briefly describe the three behavioral paradigms employed; (2) describe the construction of anatomical maps based on architectural and connectional data; (3) show the distribution of neuronal responses with respect to anatomical subdivisions; (4) validate the anatomical map with an unbiased cluster analysis of functional properties; (5) compare the neuronal responses recorded in each area at the population level.

Concerning the behavioral paradigms, we employed two "passive" tasks aimed at assessing how VLPF neurons encode visual stimuli, in the absence of a specific request to use them, and a Visuo-Motor task investigating functions related to action organization (see Materials and methods; [36,45–47]). In particular, in the Picture task (Fig 1A), the monkeys simply had to observe one of 12 pictures depicting faces, geometric solids, food or furniture; in the Video task (Fig 1B) they were required to observe one of six videos showing biological movements either goal related or not and object motion; in the Visuo-Motor task (Fig 1C), monkeys were instructed by two visual cues that they will have to either act on one of three presented objects (Action condition) or refrain from acting (Inaction condition). Then, when the object became visible, the monkeys had to wait for a go signal to perform the instructed response.

In order to define anatomical areas in the recorded brains, we relied on previous architectonical and connectional studies from our lab [20–22,40]. In those studies, we had first defined the architectonic borders of areas 8FEF, 8r, 45A, 45B, 46v, and 12r and, subsequently, we injected neural tracers in these areas on 12 monkeys (see S1 Table); the different pattern of connections observed after each injection confirmed the architectonic borders and allowed to further subdivide areas 46v and 12r in three additional sectors. Here, we superimposed the areas and sectors identified in the above-mentioned studies onto the histological reconstructions of the two recorded brains, by warping both the architectonic maps and the locations of the various injection sites used to characterize the connectivity of each area (see Materials and methods and [48]. Fig 1D and 1E depict the results of this process. In the present work, we excluded from analysis the caudal oculomotor areas 8 and 45B and pooled together the rostral portions of 46v and 12r (hereafter defined as the *rostral sector*), since the position of the recording chambers allowed us to record only a small number of sites, especially in M2 (see Materials and methods).

Within the investigated region, we recorded neural activity from 99 penetrations in M1 (Fig 1F) and 64 in M2 (Fig 1G). Note that, while the central part of the recorded region was densely and homogeneously explored in both monkeys, area 45A was more sampled in M1. Furthermore, in M2, the sampling of the rostral sector was limited to its caudalmost part.

### Distribution of neural responses recorded in the Picture task

We recorded a total of 2,289 neurons in the Picture task, and analyzed their responses by means of a 2×12 ANOVA with factors Epoch (Baseline and Presentation) and Stimuli (the 12 presented images) ($p < 0.01$, see Materials and methods) followed by Newman–Keuls post-hoc tests, which allowed us to identify neurons responding to the observation of static images and assess their possible selectivity (for the selection criteria, see Materials and methods and [46]).

                                                      

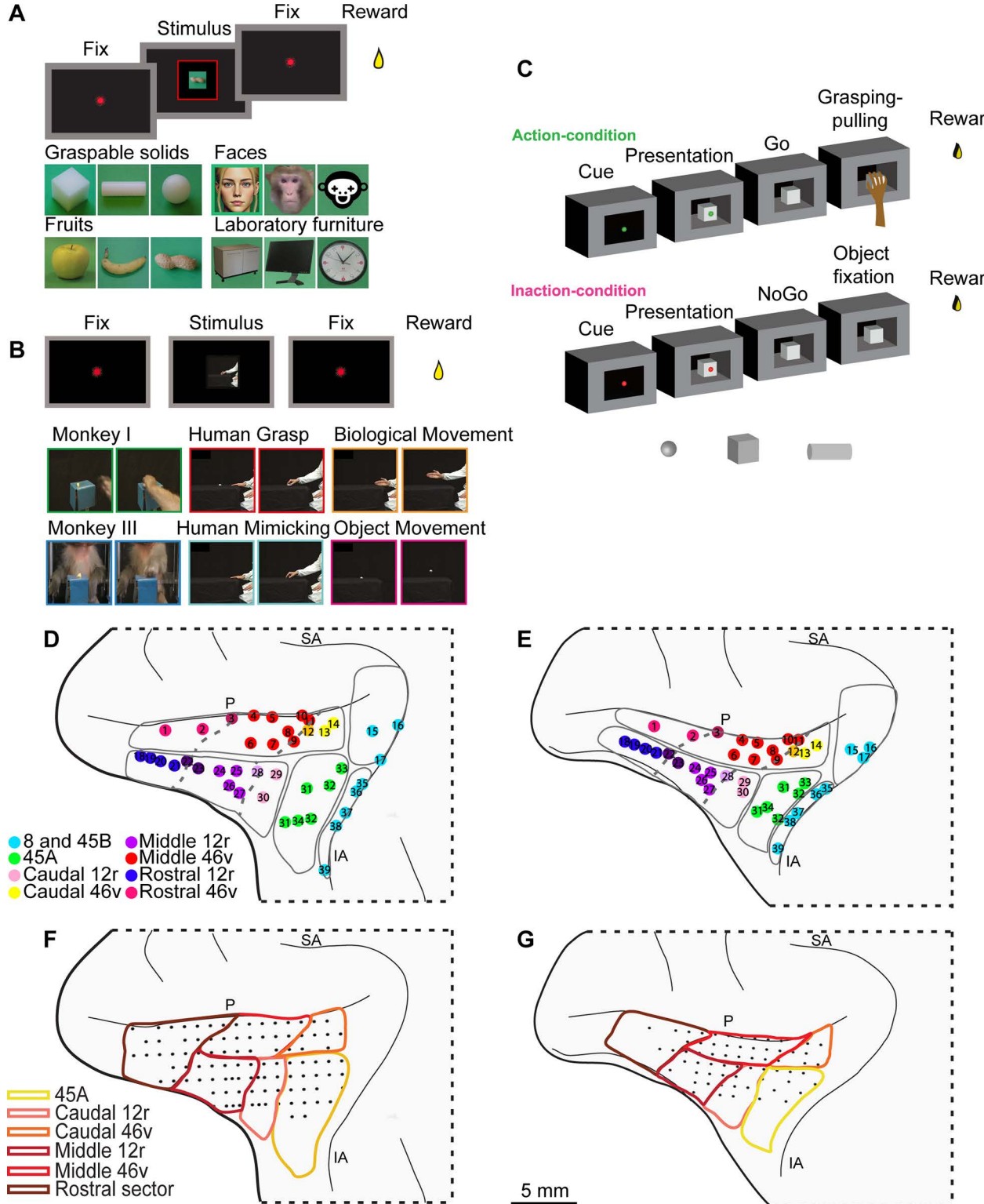

**Fig 1. Behavioral paradigms and parcellation of the recorded regions. (A and B)** Temporal sequence of events (above) and stimuli (below) of Picture and Video tasks. For privacy reasons, the image of the human face used in the experiment has been replaced in the figure with a schematic representation **(C)** Temporal sequence of events of the Visuo-Motor task. In the bottom, the three used objects are shown. **(D and E)** Localization of

the injection sites taken from the literature (see Materials and methods and S1 Table), superimposed on the cytoarchitectonically defined VLPF areas of the recorded monkeys. Each dot, labeled with an index number, represents a different injection site (see S1 Table for details on each labeled site). Each color refers to a specific pattern of connectivity characterizing the injection sites [20–22,40]. Orange, light violet, dark violet and dark magenta dots refer to territories showing transitional connectivity. Dashed lines correspond to the borders between connectionally defined areas. **(F and G)** Recording grids of the two monkeys superimposed on the anatomical parcellation shown in (D) and **(E)**. P: Principal sulcus; IA: Inferior arcuate sulcus; SA superior arcuate sulcus.

Neurons responding to the presentation of static images were well represented in the whole recorded region in both monkeys, except for area 45A, characterized by a lower number of responses in M1 than in M2 (Fig 2A and 2B).

Selective neurons were mostly located in caudal 46v, caudal 12r, and middle 12r (Fig 2C and 2D). Fig 2E depicts the response of a caudal 46v non-selective neuron (2×12 ANOVA epoch effect $p < 0.01$; interaction effect: n.s.). Fig 2F shows the response of a caudal 12r selective neuron, active only during the observation of the sphere (2×12 ANOVA interaction effect followed by Newman–Keuls post-hoc test, $p < 0.01$).

### Distribution of neural responses recorded in the Video task

We recorded a total of 2,687 neurons in the Video task, and analyzed their responses by means of a 3×6 ANOVA with factors Epoch (Baseline, Video Epoch I and II) and Stimuli (the six presented videos) ($p < 0.01$, see Materials and methods), followed by Newman–Keuls post-hoc tests, which allowed us to identify neurons responding to the observation of the videos and assess their possible selectivity according to the criteria defined in the Materials and methods section (see also [45]).

Fig 3A and 3B show the distribution of neurons responding to the observation of videos. In both monkeys, these neurons were more concentrated in the caudal and ventral parts of the recorded region. Neurons showing a preference for at least one stimulus had a similar pattern of distribution, being mostly located in the caudal and ventral areas (Fig 3C and 3D), with a larger representation, in M2, in caudal 46v.

Fig 3E shows an example of a caudal 12r non-selective neuron responding to video presentation, independent of the observed stimulus (3×6 ANOVA epoch effect, $p < 0.01$; interaction effect: n.s.). Fig 3F depicts the activity of a caudal 12r selective neuron discharging only during the observation of a monkey grasping food, seen from a third person perspective, and exclusively in the second video epoch (3×6 ANOVA interaction effect followed by Newman–Keuls post-hoc test, $p < 0.01$).

### Distribution of neural responses recorded in the Visuo-Motor task

Using the Visuo-Motor task, we recorded a total of 3,045 neurons. We analyzed the neuronal responses by means of a 9×2 ANOVA with factors Epoch (Baseline, Pre-Cue, Cue, Pre-Presentation, Presentation, Set, Go/NoGo, Grasping/Fixation, Reward) and Condition (Action and Inaction) ($p < 0.01$, see Materials and methods), followed by Newman–Keuls post-hoc tests, which allowed us to identify task-related neurons, and assess their possible preference for one of the two task conditions in each epoch, according to the criteria defined in the Materials and methods section (see also [36,47]). Fig 4A and 4B depict the distribution of task-related neurons in the two monkeys, that appeared to be quite homogeneously represented in the different areas.

Based on the above-mentioned statistical analysis, we identified neurons responding, in each epoch, either preferentially to one of the two conditions (Condition-dependent, see Materials and methods) or independently of the condition (Condition-independent).

*Cue-related* neurons were generally well represented, in both monkeys, in the whole recorded region with a slightly lower density in middle 46v. Note that, in M1, Condition-depended neurons tended to be more represented in the central sector (middle 46v, caudal 12r, and middle 12r), in M2 they were mostly distributed in the ventral part of the recorded region (area 45A, caudal 12r, and middle 12r). Neurons showing a preference for the Inaction condition were much more

PLOS Biology

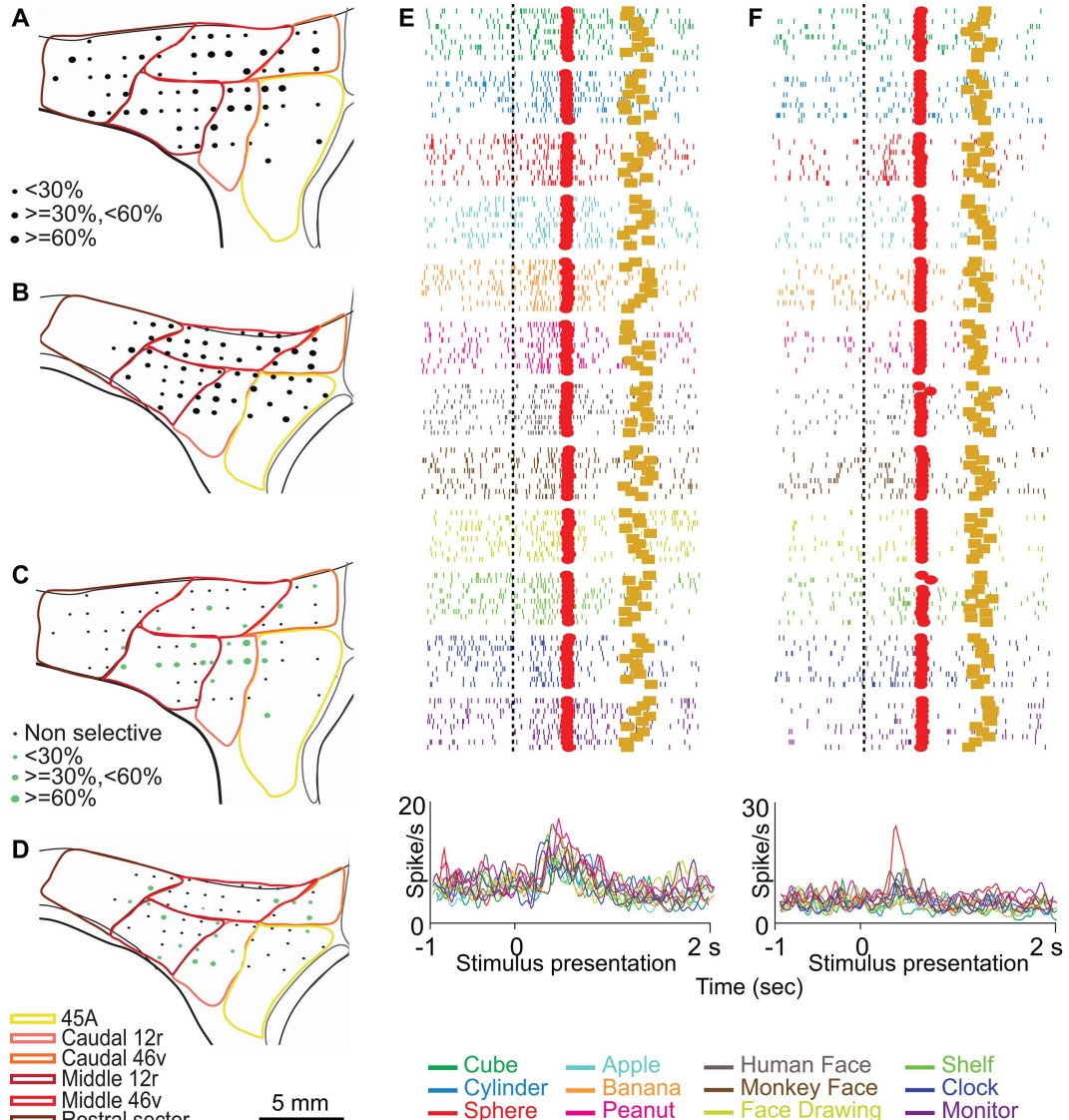

**Fig 2. Distribution of functional properties in the Picture task. (A, B)** Distribution of task-related neurons in the Picture task. The size of black dots represents the proportion of task-related neurons out of the total number of neurons of that site. **(C, D)** Distribution of selective neurons in the Picture task. The size of green dots represents, for each site, the proportion of selective neurons out of task-related neurons. **(E)** Example of neuron showing a similar discharge profile following stimulus presentation in all conditions. **(F)** Example of neuron responding selectively to the presentation of the sphere. Rasters and histograms are aligned with stimulus presentation (vertical dashed line). Red squares: stimulus offset; Gold squares: reward delivery. Abscissae: time **(s)**; Ordinates: firing rate (spikes/s).

represented than those preferring the Action condition (Fig 4C and 4D). Inaction-related neurons were less represented in caudal 46v and in the rostral sector. Fig 4E shows the discharge of a 45A Condition-dependent neuron responding to cue appearance only in the Inaction condition (9×2 ANOVA, interaction effect followed by Newman–Keuls post-hoc test, $p < 0.01$).

*Presentation-related* responses were quite homogeneously distributed over the whole recorded region in both monkeys. However, in M1 these responses were relatively less represented in middle 12r and in 45A (Fig 4F and 4G).

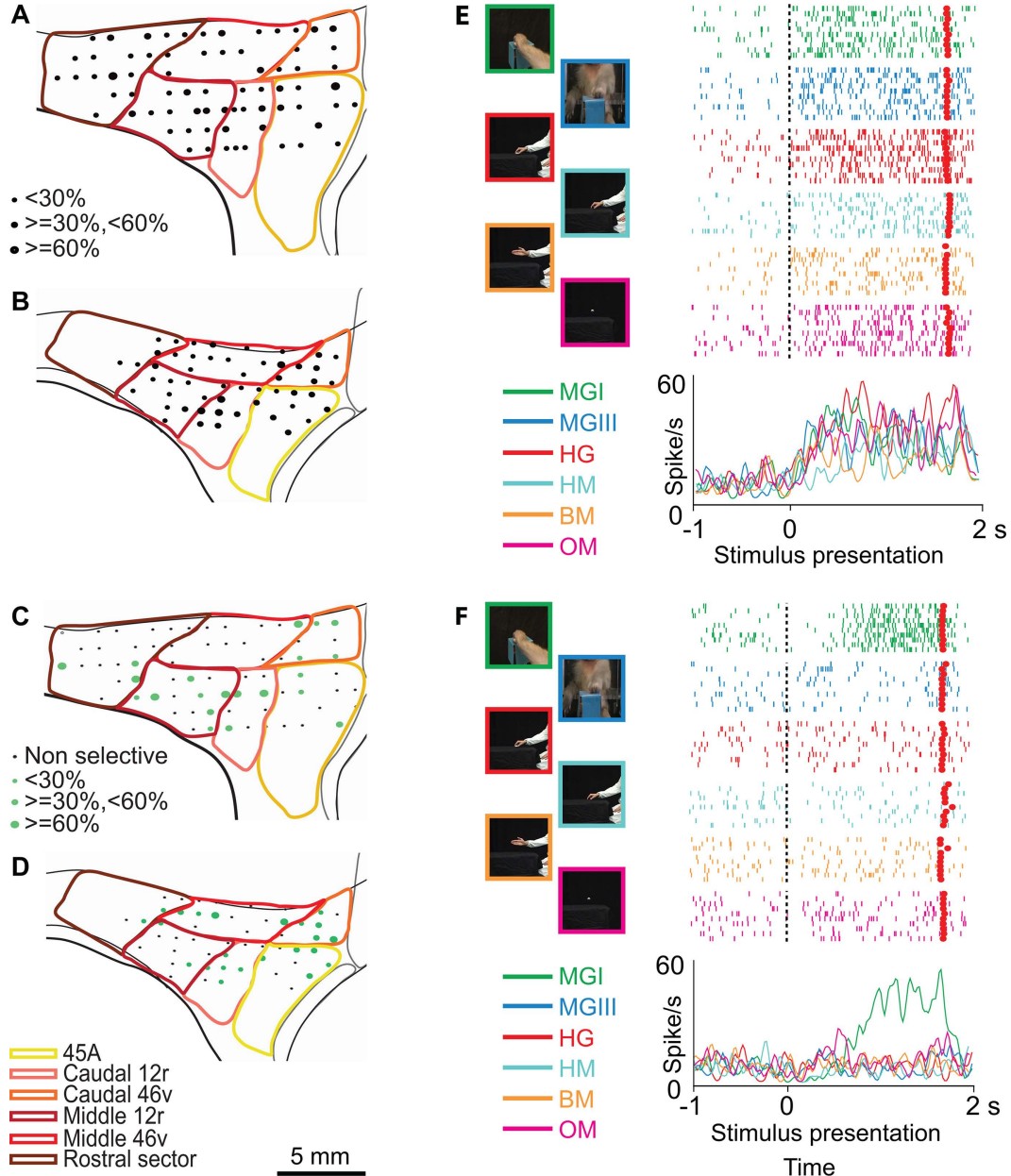

**Fig 3. Distribution of functional properties in the Video task. (A, B)** Distribution of task-related neurons in the Video task. **(C, D)** Distribution of selective neurons in the Video task. **(E)** Example of neuron showing a similar discharge profile following stimulus presentation in all conditions. **(F)** Example of neuron responding selectively to presentation of a monkey grasping in first person perspective. Other conventions as in Fig 2.

Condition-dependent neurons were unevenly distributed in both monkeys, being more densely represented in middle 46v, caudal 46v, and middle 12r in M1 and in middle 46v, caudal 12r, and middle 12r in M2. Fig 4H shows the discharge of a middle 12r Condition-dependent neuron responding to object presentation in both conditions, but significantly stronger in the Action condition (9×2 ANOVA interaction effect followed by Newman–Keuls post-hoc test, $p < 0.01$).

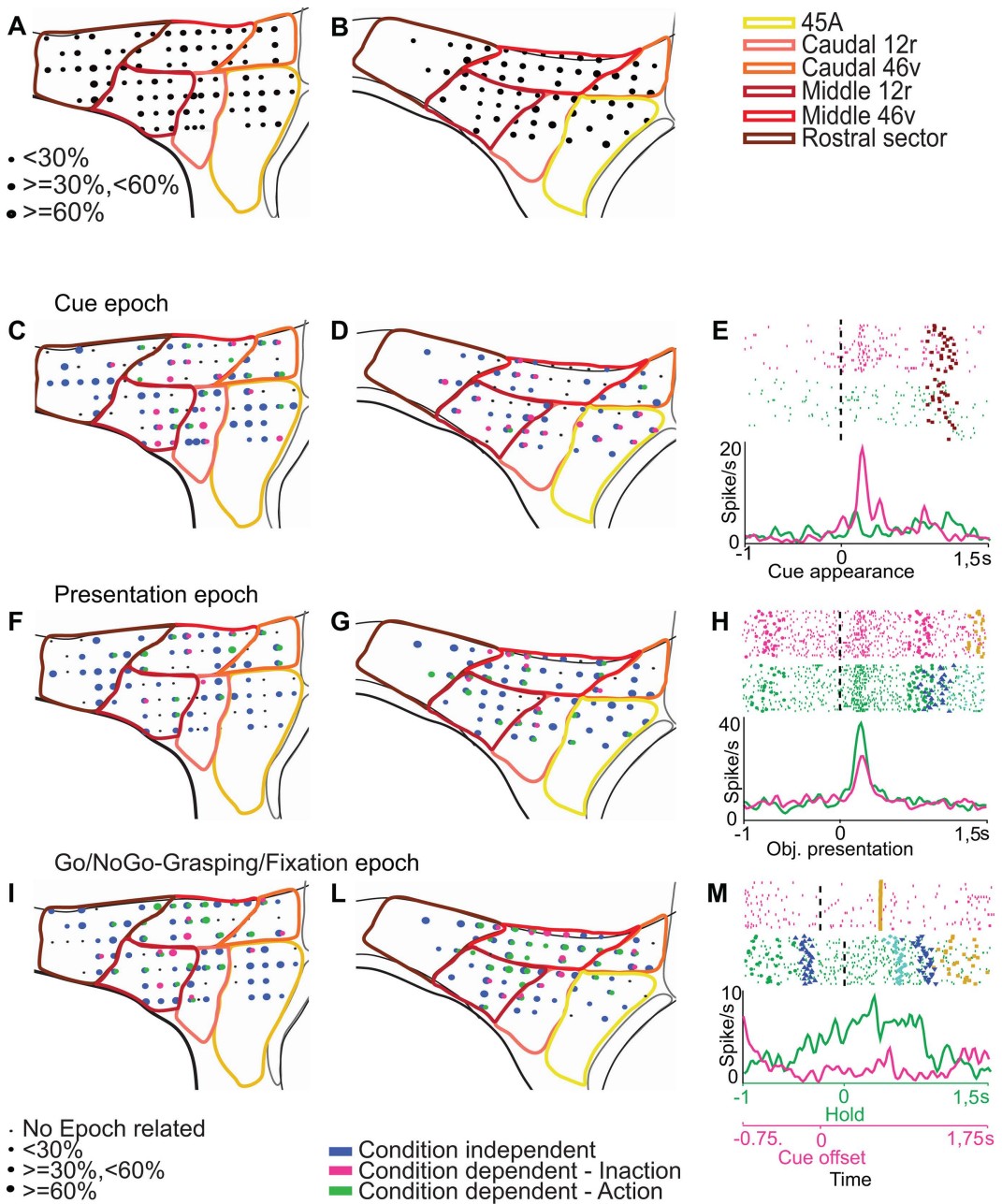

**Fig 4. Distribution of functional properties neurons in the Visuo-Motor task. (A and B)** Distribution of task-related neurons. **(C, D, F, G, I, L)** Distribution of Condition-dependent and Condition-Independent neurons responding during the Cue, Presentation, and Go/NoGo-Grasping/Fixation epochs. Black dots represent penetration sites in which no epoch-related neurons have been recorded. Blue, green, and magenta dots correspond to the sites in which Condition-independent, Action-selective or Inaction-selective epoch-related neurons, respectively, are represented. The size of colored dots represents the proportion of the respective category of epoch-related neurons on the total number of task-related neurons observed in each site. **(E)** Example of neuron showing a preferential discharge following cue appearance in the Inaction condition. **(H)** Example of neuron showing a discharge rate after object presentation, which is higher during the Action compared to the Inaction condition. **(M)** Example of neuron showing a discharge starting from the offset of the instructing cue, only during the Action condition. Rasters and histograms are aligned (vertical dashed line) with cue onset **(E)**, object presentation **(H)**, and object holding/cue offset (M, for Action and Inaction conditions, respectively). Green/Magenta circles: Action and Inaction cue appearance and offset, Brown squares: object presentation; Blue upward triangles: release of the hand from the starting position; Blue downward triangles: return of the hand on the starting position; Gold squares: reward delivery. Other conventions as in Fig 2.

Neurons responding during the last part of the task (*Go/NoGo-Grasping/Fixation* epoch) were well distributed, in both monkeys, along the whole explored region (Fig 4I and 4L), with Condition-dependent neurons being mainly located in its central part (caudal 12r, middle 12r, middle 46v, and caudal 46v), while area 45A and the rostral sector were mainly characterized by Condition-independent neurons. Fig 4M depicts the discharge of a middle 46v Condition-dependent neuron responding only in the Action condition (9×2 ANOVA interaction effect followed by Newman–Keuls post-hoc test, $p < 0.01$). The neuron starts discharging just before the beginning of movement, peaks during object pulling, and abruptly ceases firing when the action is completed with the hand returning to the starting position.

## Unsupervised clustering of task-related neurons

To verify whether the distribution of functional properties aligns with the above-described anatomical parcellation, for each task, we performed an unsupervised clustering of firing rates of task-related neurons, using their representation on the UMAP embedding (see Materials and methods). We first performed this analysis on the whole population of task-related neurons (see Materials and methods for the selection criteria) of the Picture, Video, and Visuo-Motor tasks. For each task, we identified two functional clusters. S1 Fig depicts the distribution of neurons belonging to each cluster (see Materials and methods for details on map construction). Functional clusters were distributed along the whole recorded region, thus not showing any clear distinction between areas. S1 Fig also shows the baseline-subtracted mean firing rate for the populations of neurons in each identified cluster. For each task, these neurons showed a modulation of activity time locked to the main events of the tasks. More specifically, one cluster was typically characterized by an increase in firing rate after each event, with the strongest response following stimuli/object presentation, while the other showed a strong peak of discharge aligned with the first task event/task beginning, followed by a prolonged and general decrease of the firing rate.

The same analysis was performed on neurons classified as selective in the Picture and Video tasks and as Condition-dependent in the Cue, Presentation, or Go/NoGo-Grasping/Fixation epochs of the Visuo-Motor task (see Materials and methods for the selection criteria). Differently from the above-described distribution of task-related neurons, these clusters covered smaller and more localized regions (S2 Fig). The anatomical distribution of functional clusters related to the responses to visual stimuli in the Picture and Video tasks and to the final phases of the visuo-motor task, was very consistent in the two monkeys, while that associated to condition-dependent Cue and Presentation responses showed some difference between monkeys.

This analysis allowed us to observe that, in both monkeys, functional subdivisions overlap with anatomically defined areas, confirming each anatomical border. For example, this was evident when considering middle 46v, which was characterized by a strong representation of clusters related to the final phase of the visuo-motor task and by the almost complete absence of clusters related to selective responses in the passive tasks, instead represented in the neighboring caudal and ventral areas. Another functional border can be observed between caudal and middle 12r, with clusters related to final phases of the visuo-motor task more consistently represented in the middle than in the caudal sector of area 12r, and clusters related to the appearance of visual stimuli (both in the passive and the visuo-motor tasks) tending to be more represented and overlapping in caudal 12r.

Clustering analysis confirmed, in both monkeys, our anatomical parcellation of the VLPF, thus allowing us to analyze and compare the population data related to the identified anatomo-functional areas, considering the two monkeys together.

## Representation of task-relevant factors in VLPF areas

To evaluate how the population activity of the different VLPF areas is modulated by task relevant factors, for each task we performed demixed principal component analysis (dPCA, see Materials and methods) on task-related and on all recorded neurons. This analysis allowed us to identify those components explaining variability that is related to the factors considered in each task, as well as factor-independent variability. These components were then used to define low-dimensional

subspaces that isolate the neural dynamics associated with each experimental factor. The resulting dynamics are represented by the neural trajectories shown in the figures described in the following paragraphs (see [49] for a similar approach, and Materials and methods for further details).

**Picture task.** Figs 5 and S3 show that the trajectories of each area were characterized by peaks linked to stimulus presentation, represented by deflections along the Factor-independent axis. The distinction between trajectories representing the different stimuli (Type of stimulus axis), is very evident only in caudal 12r.

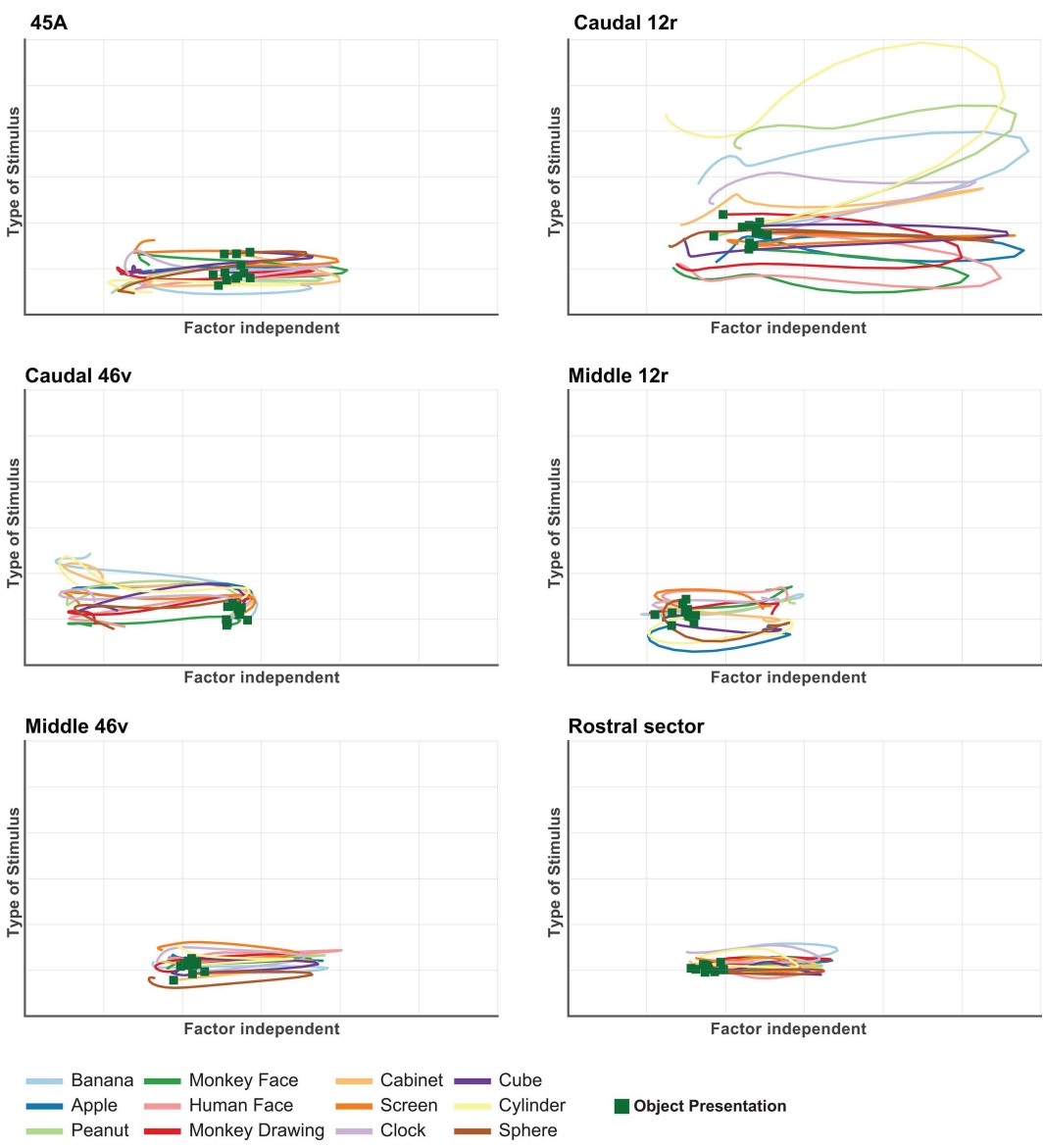

**Fig 5. dPCA trajectories of each VLPF area in the Picture task.** Each panel depicts the time course of the first Factor-Independent (*X* axis) and of the first Type of Stimulus-related (*Y* axis) principal components plotted together, relative to the population of task-related neurons recorded in each area. Each colored line corresponds to one of the 12 stimuli presented in the task. Green squares represent the time at which the stimulus presentation occurs. The data matrices (see Materials and methods) underlying this figure can be found in OFS database: https://osf.io/j8fcs/.

**Video task.** Figs 6 and S4 show that the trajectories of each area were characterized by peaks linked to video presentation, represented by deflections along the Factor-independent axis. The distinction between trajectories representing the different videos (Type of stimulus axis), is very evident in caudal areas (45a, caudal 12r, and caudal 46v) and in middle 12r. Note that in caudal 12r and caudal 46v, the trajectories representing stimuli belonging to similar categories [45] are grouped together.

**Visuo-motor task.** Figs 7 and S5 show that the trajectories of each area are characterized by peaks linked to the main task events (e.g., cue onset, object presentation, cue offset), represented by deflections along the Factor-independent axis. In each area, trajectories were clearly separated on the Condition axis (Action and Inaction). However, this

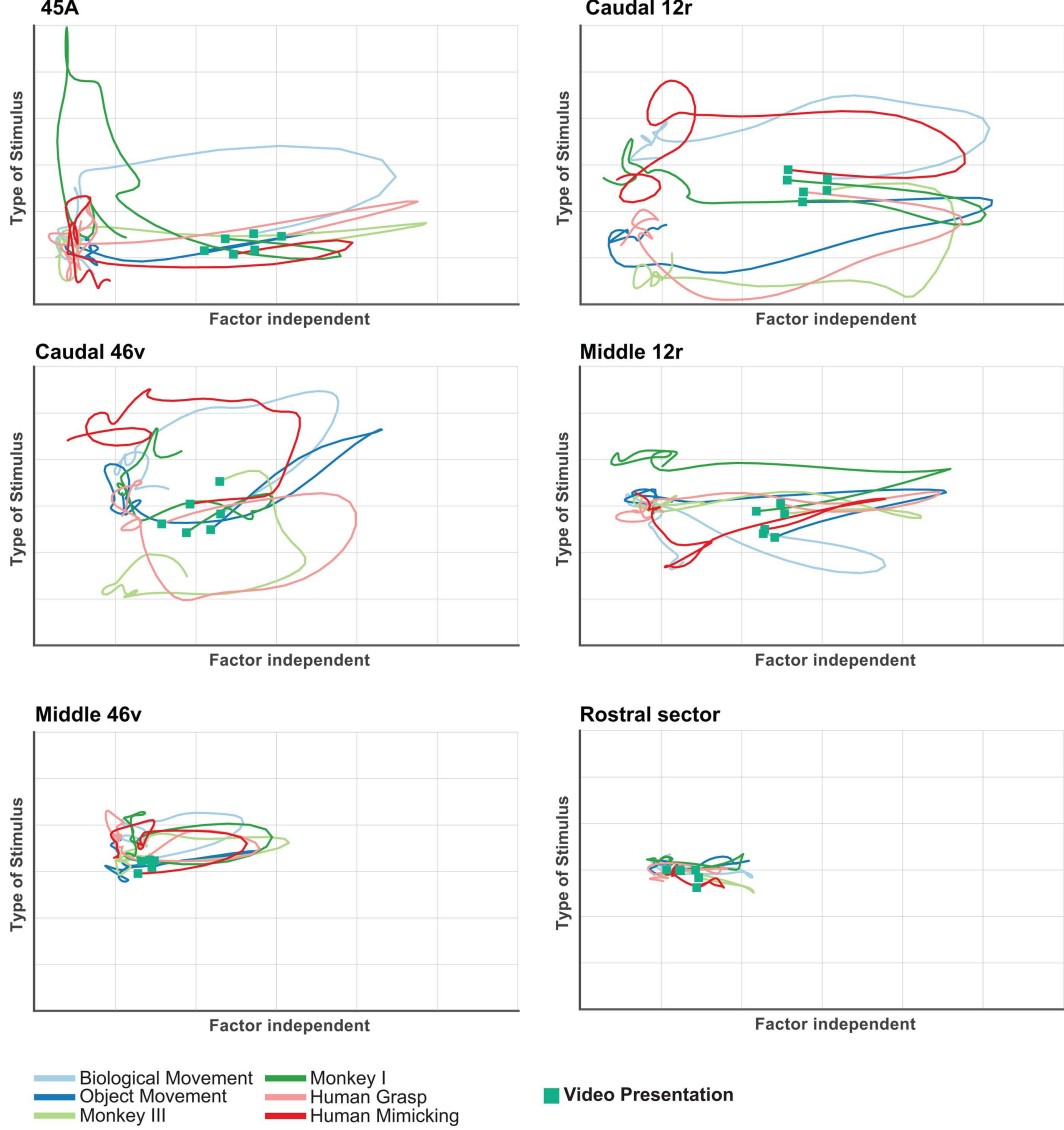

**Fig 6. dPCA trajectories of each VLPF area in the Video task.** Each panel depicts the time course of the first Factor-Independent (X axis) and of the first Type of Stimulus-related (Y axis) principal components plotted together, relative to the population of task-related neurons recorded in each area. Each colored line corresponds to one of the six videos presented in the task. Green squares represent the time at which the stimulus presentation occurs. The data matrices (see Materials and methods) underlying this figure can be found in OFS database: https://osf.io/j8fcs/.

separation is less evident in 45A and the rostral sector. Concerning the other areas, this segregation occurred just after cue appearance in caudal 46v and middle 12r, and later, between cue and object presentation, in middle 46v, whereas this separation occurred only after object presentation in caudal 12r. Moreover, trajectories maintained their separation after the cue offset in middle 12r and middle 46v, while in caudal 12r and caudal 46v, in this period, they converged toward their origin.

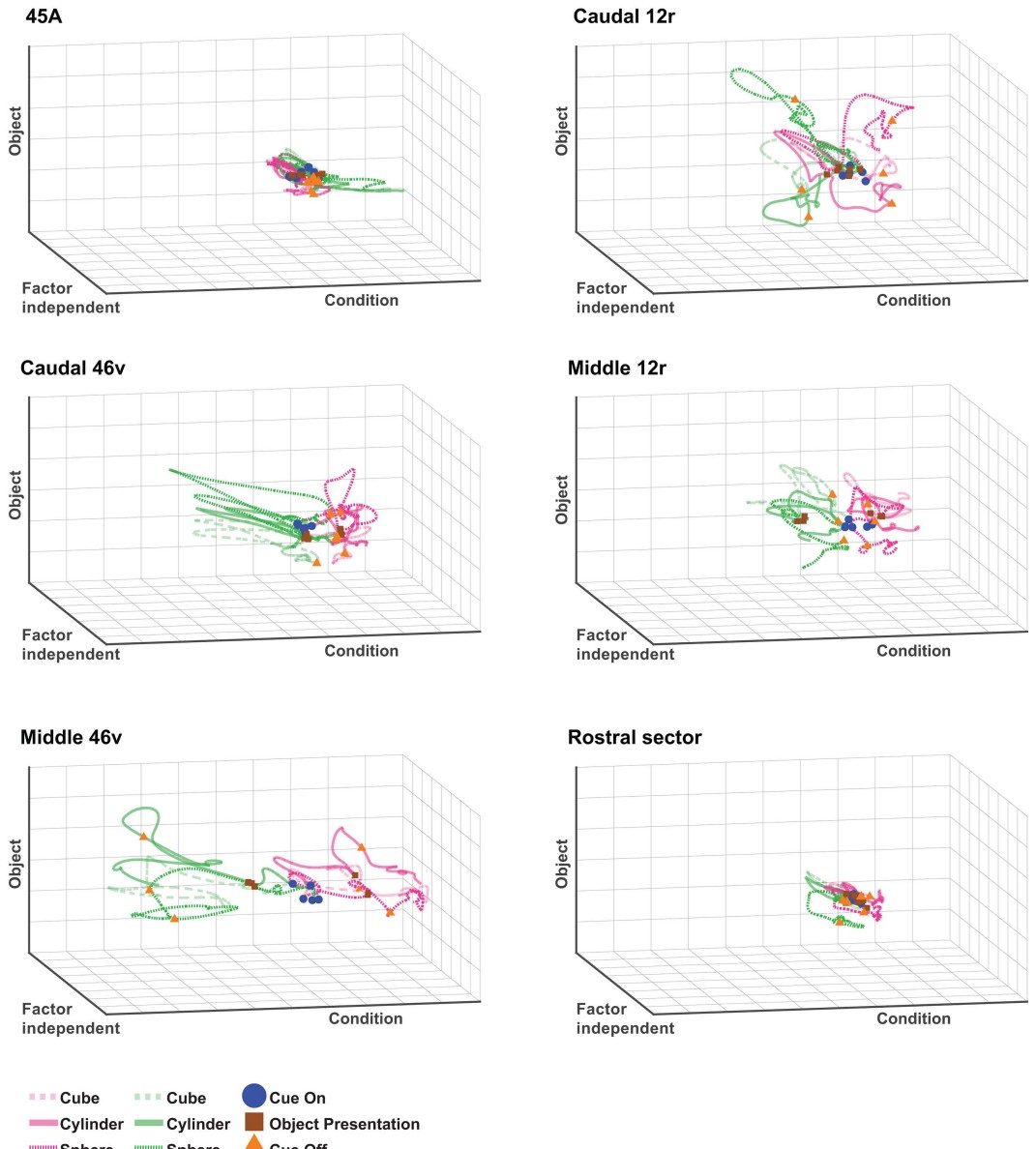

**Fig 7. dPCA trajectories of each VLPF area in the Visuo-Motor task.** Each panel depicts the time course of the first Condition-related (*X* axis), of the first Object-related (*Y* axis) and of the first Factor-Independent (*Z* axis) principal components, relative to the population of task-related neurons recorded in each area. Green and Magenta colored lines represent Action-related and Inaction-related trajectories, respectively. Continuous and dashed lines correspond to the three objects presented, respectively. Blue circles, brown squares, and orange triangles represent, for each trajectory, the time of cue onset, object presentation, and cue offset, respectively. The data matrices (see Materials and methods) underlying this figure can be found in OFS database: https://osf.io/j8fcs/.

Concerning the Object axis (Cube, Cylinder, and Sphere), all areas except 45A and rostral sector showed a separation of the trajectories after object presentation. Interestingly, object-related separation was apparently more evident in the Action than in the Inaction condition.

## Population activity of VLPF areas in the Visuo-Motor task

In order to evaluate the response of the population of neurons recorded in the different VLPF areas, we plotted, for each of them, the baseline-subtracted activity (see Materials and methods) of task-related neurons (Fig 8) and of all recorded neurons (S6 Fig) in the Action and Inaction conditions.

Both analyses showed that all areas share, as a common feature, the presence of two clear peaks following cue and object appearance. The peak related to cue presentation during the Inaction condition was higher than the one observed during the Action condition in all areas, except for caudal 46v, where this pattern is inverted. The object presentation-related peak was, in all areas, higher than the cue-related one, with the Action condition always eliciting a stronger neural population activity than the Inaction one during this period.

The pattern of responses observed in the final part of the task (starting around the Go/NoGo signal) was quite heterogeneous when comparing the different areas (Figs 8 and S6). Some areas were primarily characterized by an enhanced population response, while others displayed a suppressed response (i.e., below baseline level). In particular, the rostral sector exhibited suppressed population activity in both Action and Inaction conditions, while 45A showed activity suppression only in the Inaction condition. Caudal 46v demonstrated slight suppression of activity, mainly occurring in the Action condition just before the Go/NoGo signal. Note that when considering the population of task-related neurons (Fig 8), the activity suppression observed in this condition is also maintained during object holding. Caudal 12r population activity began to increase in both conditions following the Go/NoGo signal, reaching a peak after about 250 ms. In the Inaction condition, the activity remained sustained until reward delivery, then sharply falling below baseline level. In contrast, in the Action condition, sustained activity ended with the beginning of holding, abruptly dropping to baseline level during this period, and decreasing below it after reward delivery. Middle 12r was characterized by a sustained increase in firing rate only during the Action condition. The population response began to grow after the Go signal, reaching a peak around the start of the holding. Then, the activity decreased to baseline level, while the monkeys continued pulling the object. In both conditions, there was a final lower peak after reward delivery. Middle 46v exhibited the highest mean-net activity among all the areas considered in the final task segment of the Action condition. Notably, while no clear enhancement or suppression was evident in the Inaction condition, in the Action condition the activity increased after the Go signal, continued to rise during reaching and grasping execution, peaked around the start of holding, and gradually decreased while the monkeys held the object, returning to baseline level around reward delivery.

## Decoding of the condition and object factors in VLPF areas

We adopted a cross-temporal decoding analysis on the neural activity recorded in the Visuo-Motor task to assess, in a time-detailed manner, which type of information was encoded by the studied areas. In particular, this approach consisted in training a classifier in decoding the Condition or Object on a specific time point and testing it on the same or on all others (see Materials and methods for details). This allowed us to assess whether a high and significant level of accuracy was achieved not only when training and testing the classifier on the same time bin (on-diagonal time points), but also when the training and testing phases involved different periods (off-diagonal time points; see Materials and methods for details). We refer to the latter case as a 'static pattern' of coding, indicating that a similar neural code, associated to the analyzed factor (Condition or Object) is maintained across the training and testing periods.

Figs 9 and S7 show the accuracy level of the decoding of the Condition (A–F) and the Object (A′–F′) factors observed in both on and off-diagonal time points (see Materials and methods). Similarly to what was described above, we observed,

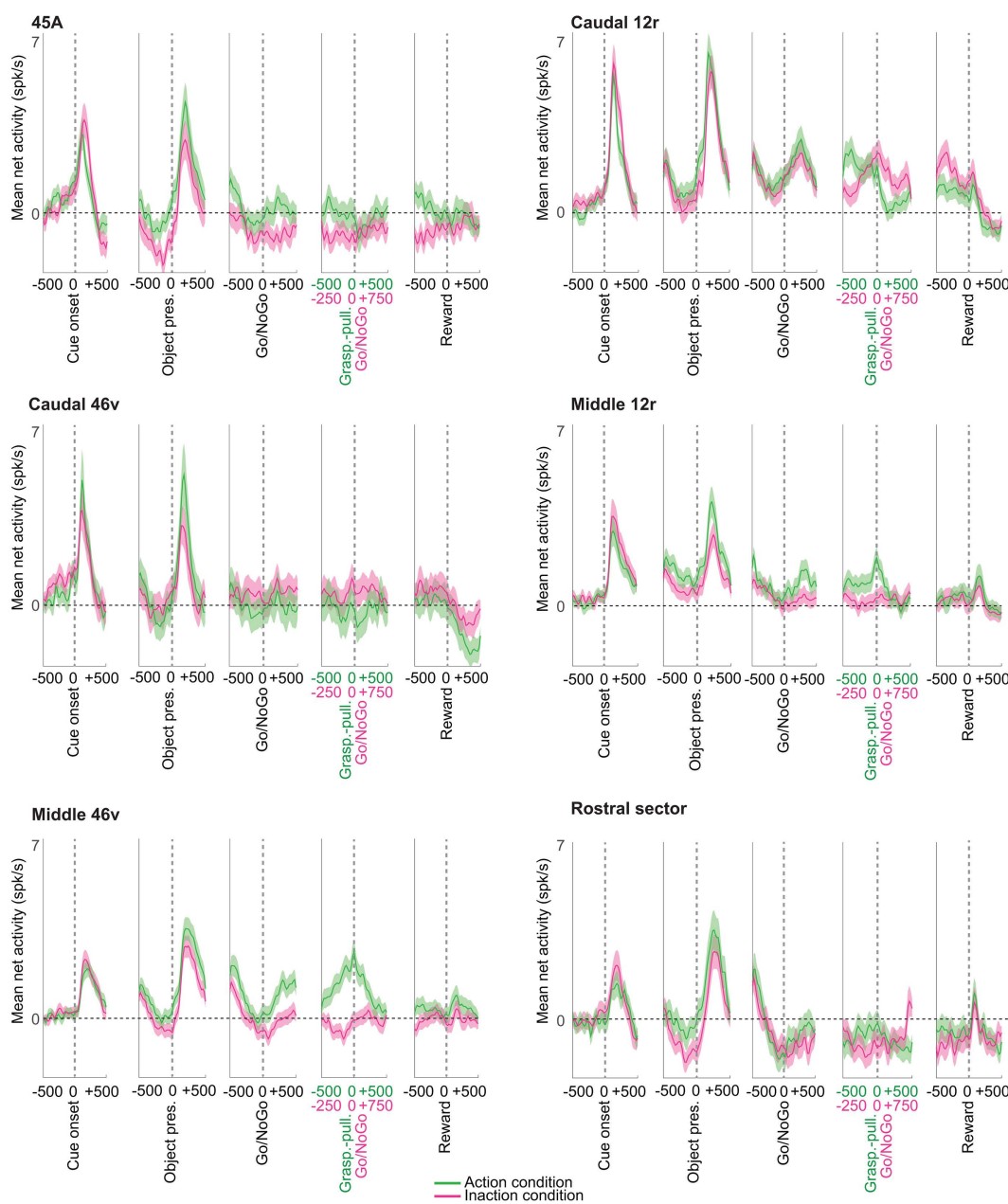

**Fig 8. Mean population activity of neurons recorded in each area during the Visuo-Motor task.** Temporal profile of mean net activity of the population of task-related neurons recorded in each of the considered VLPF areas. The magenta and green curves indicate the population mean net activity in the Inaction and Action conditions, respectively. The shaded area around each curve represents standard errors. The dashed line indicates baseline-level activity. The neuronal activity is aligned on the main task events indicated below each panel of the figure. The data matrices (see Materials and methods) underlying this figure can be found in OFS database: https://osf.io/j8fcs/.

for each area, strong consistency between the results based on task-related and the whole population of recorded neurons, thus, we will describe them together (Figs 9 and S7).

The Cross-temporal decoding analysis showed that some functional features are shared by all the investigated areas, while others characterize each of them. Considering the Condition factor, in each area, a very high and significant level of accuracy was evident on the diagonal, when the classifier was trained and tested on the same time bins (permutation test $p < 9.5e{-6}$, Bonferroni corrected for the number of on-diagonal time bins, see Materials and methods), with a relative decrease in accuracy typically occurring around the Decision period.

Besides the important similarities among areas, which are evident when considering the accuracy levels along the diagonal, static patterns involving different task periods characterized each area. In area 45A, a static pattern of coding was observed between the final phase of the Cue period and the final part of the Behavioral response period (cluster-based permutation test, $p < 0.001$, see Figs 9 and S7 and Materials and methods). Note that this pattern was "bi-directional", being present both when training on the Cue period and testing on the Behavioral response one and vice versa. In caudal 12r, a static pattern of activity was observed among the Presentation, Decision, and Behavioral response periods (cluster-based permutation test, $p < 0.001$; Figs 9 and S7), being equally evident in both decoding "directions". A static pattern of coding was instead observed, in caudal 46v, across various time periods: Cue, Presentation, and a phase extending from the end of the Decision period to the beginning of Behavioral response one (cluster-based permutation test, $p < 0.001$; Figs 9 and S7). Note that the static pattern observed in this area was not clearly "bi-directional", being mostly evident when training the classifier on the last part of the task and testing it on the Cue and Presentation periods. Considering middle 12r, cross-temporal decoding of the Condition factor revealed that a high and significant level of accuracy was present when training the algorithm on the Decision period and testing it on the Cue and Behavioral response period (cluster-based permutation test, $p < 0.001$; Figs 9 and S7). In middle 46v, a very consistent and "bi-directional" static pattern of coding occurred during an extended period of the task, encompassing the Presentation, Decision and Behavioral response periods (cluster-based permutation test, $p < 0.001$; Figs 9 and S7). Finally, in the rostral sector, a static pattern of coding, though associated with a quite low level of accuracy, was observed only when training the classifier on the Decision period and testing it on the Behavioral response one, and vice versa (cluster-based permutation test, $p < 0.001$; Figs 9 and S7).

Taking into account the Object factor, a very high and significant accuracy was always observed, in each area, along the diagonal during the Presentation period (permutation test $p < 9.5e{-6}$, Bonferroni corrected for the number of on-diagonal time bins, see Materials and methods). Note that this effect was mostly limited to this period in areas 45A, middle 46v, and in the rostral sector (Figs 9 and S7), while caudal 46v, caudal 12r, and middle 12r consistently showed very high and significant levels of accuracy, along the diagonal, from the Presentation period onwards. In addition, these latter areas were also characterized by a "bi-directional" static pattern of coding that was observed from the final phase of the Presentation period onwards (cluster-based permutation test, $p < 0.001$), and that was especially evident in caudal 12r.

## Discussion

In the present study, we investigated the specific contribution of different VLPF areas in processing sensory information and exploiting it for guiding behavior. To this aim, we recorded neuronal activity from a large sector of monkey VLPF, covering about its posterior two-thirds, during tasks involving either the passive processing of visual stimuli or their use for guiding reaching-grasping actions.

Under the hypothesis that the connectional features of different areas underpin functional specificity, we adopted an anatomical parcellation produced by our group [20–22,40] and validated it on a functional ground by using an unsupervised clustering analysis. Then, we analyzed the neuronal activity of each identified anatomical area showing that some functional features are broadly distributed, while others characterize each area.

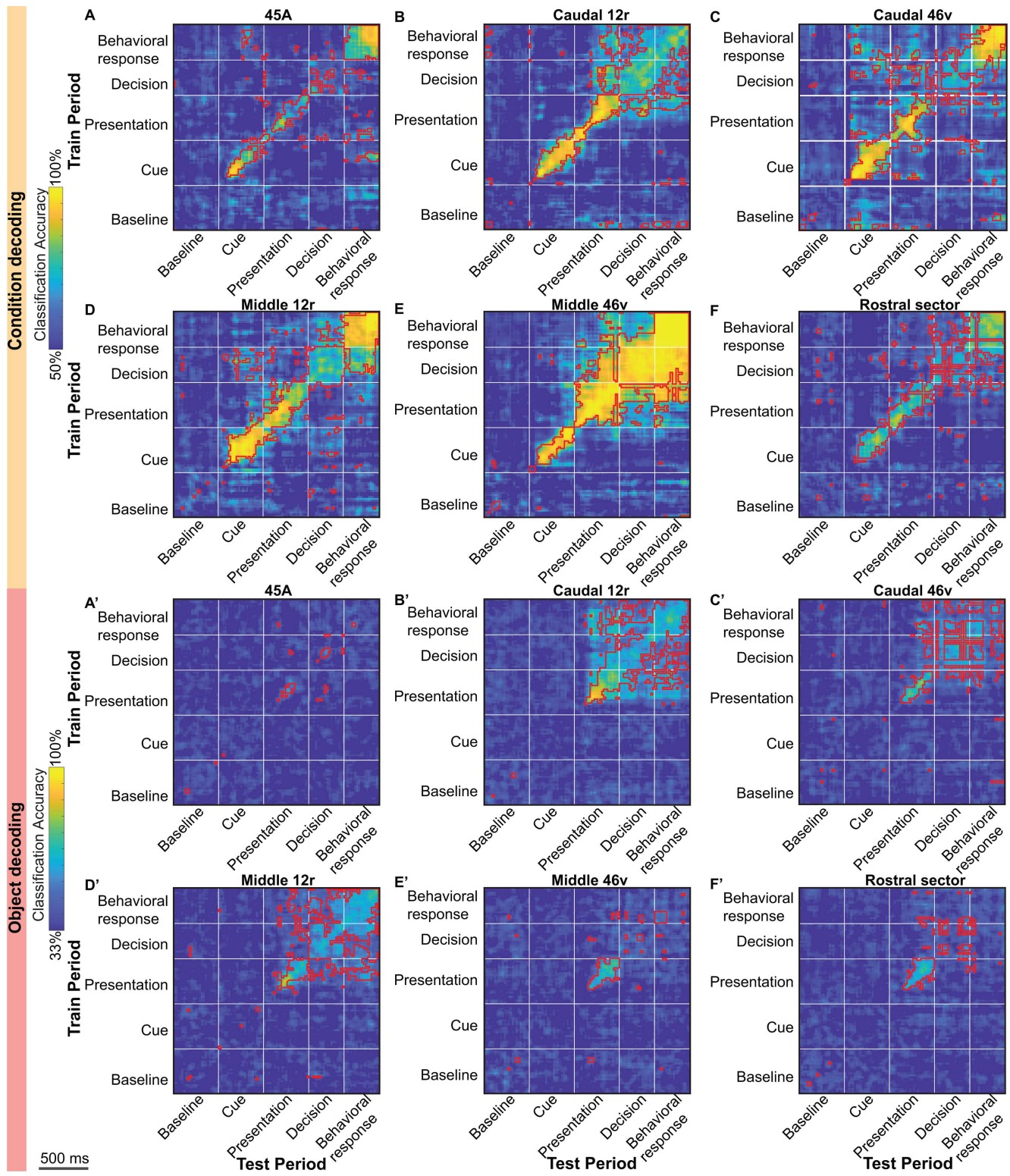

**Fig 9. Cross-temporal decoding of the Condition (A–F) and Object (A′–F′) factors of the Visuo-Motor task in task-related neurons.** For each analysis, the decoding accuracy is computed in bins of 60 ms, sampled at 20 ms intervals. For each plot, the white vertical and horizontal lines delimit the considered time periods (see Materials and methods). Decoding periods of testing and training are indicated on the X and Y axes, respectively. For

each plot, red outlines delimit statistically significant time bins (on and off-diagonal, see Materials and methods). The data matrices (see Materials and methods) underlying this figure can be found in OFS database: https://osf.io/j8fcs/.

Our main results show that:

- The coding of different task phases and the behavioral rule is observed across all recorded areas, indicating a distributed representation of these processes within VLPF, which is in line with the strong interconnection among prefrontal areas [20,22];

- The passive presentation of visual stimuli primarily activates neurons in the caudal VLPF areas, especially in caudal 12r. This suggests that these areas, consistently with their strong connections with the inferotemporal cortex [20], represent the first VLPF stage of visual processing;

  - The actual execution or withholding of grasping actions predominantly activates neurons in the intermediate VLPF areas, particularly in middle 46v. This indicates that these areas, in line with their strong connections with the parietal and premotor cortices [22], may primarily contribute to action selection and guidance;• Each area differently contributes to the encoding of visual features and to the exploitation of contextual information for guiding behavior.

  A summary view of our findings is shown in Fig 10.

## Comparison with previous mapping studies

In the past, several studies tried to functionally subdivide LPF by using visual stimuli either passively presented or in a context of working memory tasks [14,16,26,50]; however, these tasks were usually examined in isolation from one another. An exception is represented by the work of Tanila and colleagues, who studied neural activity in a large peri-principal region during the presentation of visual, auditory, and somatosensory stimuli, as well as during the execution of hand and eye movements in a naturalistic context [34]. Interestingly, they found that visual responses were most densely represented in caudal VLPF, where also oculomotor neurons have been recorded, and somatic responses (i.e., those to somatosensory stimulation and/or associated to the execution of reaching-grasping actions) were primarily found in intermediate VLPF. Accordingly, they suggested the presence of a functional distinction among different prefrontal sectors, based on their connectional properties. Indeed, they matched the distribution of single neurons characterized by specific functional properties with an anatomical framework based on connectional studies in which large injections, encompassing more than one cytoarchitectonic area, were performed [51–56]. Similarly, in our study, we started from the hypothesis that each anatomical area is characterized by specific functions depending on its pattern of connections. However, differently from Tanila and coworkers, we employed as a reference framework a more refined multi-architectonic and connectional parcellation from our group, then we functionally validated it using an unsupervised clustering approach and, finally, used this parcellation to verify whether different areas have different functions. Another difference with Tanila's study is that, in order to study single neurons responses, we used well-controlled paradigms. Thus, our study on the one hand confirms the pioneering observations of Tanila and coworkers, on the other, identifies the specific functional role in visual processing and context-based behavioral organization of the different VLPF areas.

More recently, a mapping of visual responses in LPF has been described in a series of studies performed by the group of Constantinidis [14,15,27]. Although it is difficult to directly compare our data with their findings, mainly due to the use of markedly different anatomical frameworks, some clear similarities can be found. Indeed, in a recent study [15], the authors compared the responses to passive presentation of visual stimuli with those elicited by the same stimuli when used in a working memory task. They demonstrated that posterior regions are typically more involved than anterior ones in coding visual stimuli during stimulus presentation and delay phases of the tasks, independently of whether the task was passive or memory related, while anterior areas are typically more involved in the memory task than in the passive one [15]. In

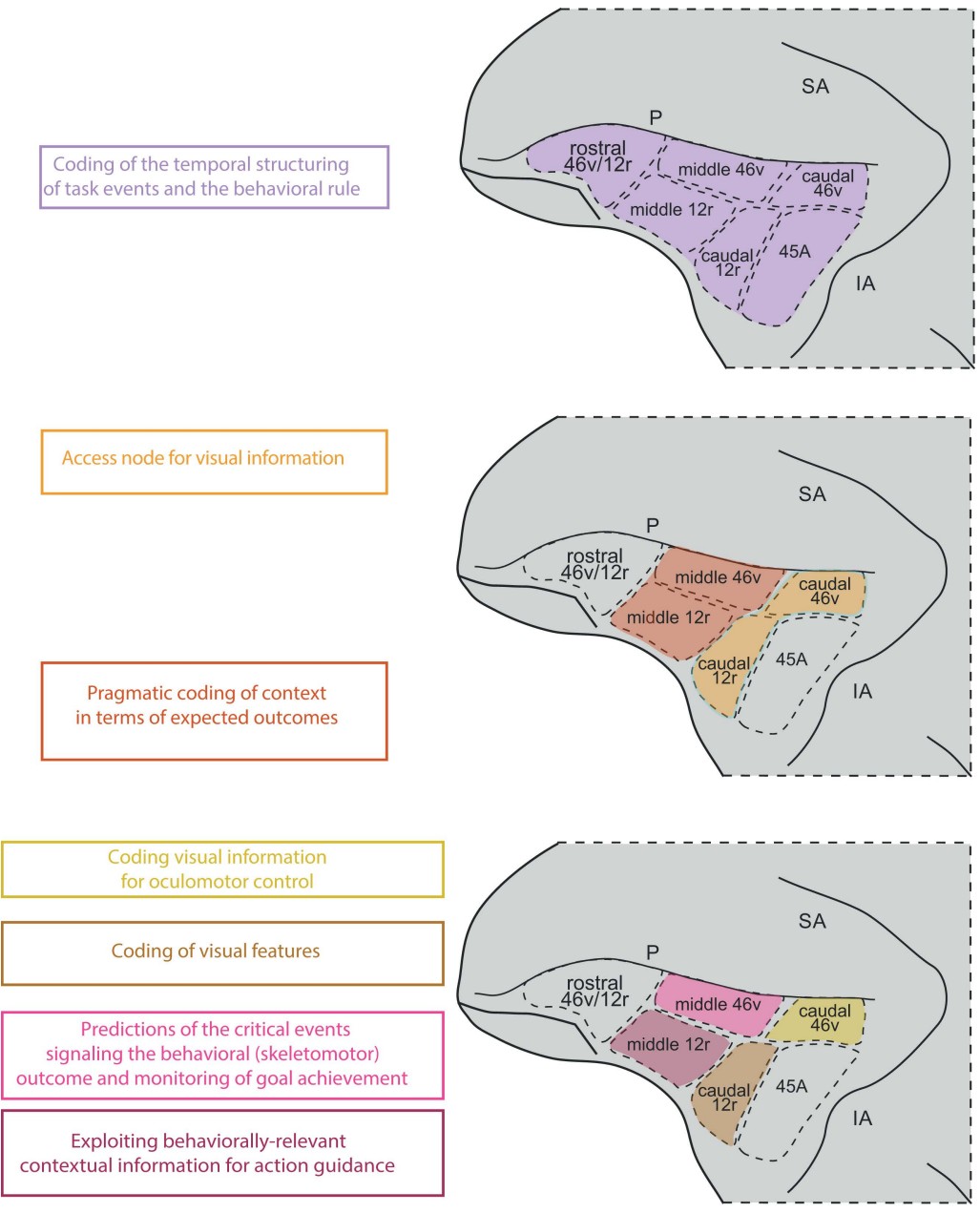

**Fig 10. Overview of the distribution of functional properties in VLPF.** P, Principal sulcus; IA, inferior arcuate sulcus; SA, superior arcuate sulcus.

other words, these authors described the presence of a rostro-caudal functional gradient within the prefrontal cortex, in which posterior areas are more involved in coding visual information per se, while more anterior regions have a stronger role in exploiting visual information to guide behavior. Our results generally confirmed this type of functional organization, allowed us to extend this idea from working memory to action organization and showed that each area gives a partly different contribution within this gradient.

## Coding of task phases and behavioral rules is broadly distributed in VLPF

Single neuron analyses reveal that task-related neurons recorded in the different behavioral paradigms are broadly represented along the whole investigated VLPF region (Figs 2–4A and 4B). Considering the results of the analyses at the population level, it emerges that all areas show multiple peaks of activity related to the main task events, indicating that they are all involved in coding their temporal structuring (Figs 7, 8, and S1). Another shared function corresponds to the coding of the behavioral rule as shown by the decoding and principal component analyses on the Visuo-motor task (Figs 7 and 9). Altogether, these results are in line with a large body of literature on the functional role of prefrontal neurons [2,9,57–59].

Note that our data fit well theoretical models describing the adaptability of functional properties in the prefrontal cortex [42–44], which propose that, although neurons encoding different types of information are broadly distributed, different subregions are characterized by a maximal sensitivity to a specific type of information. These models also propose that the complexity of the paradigms and the level of analysis might influence the possibility to describe a broader distribution of properties or the presence of regional specialization. Our task design, and the employed analyses allowed us to demonstrate that specific and distributed functions coexist in VLPF.

## The posterior region of VLPF constitutes an access node for visual information

The data from our visual tasks show that caudal and ventral VLPF areas are strongly involved in processing passively presented visual stimuli, independently of any association with a behavioral outcome. More specifically, our results show that selective responses to passive stimuli are strongly represented in caudal 12r, caudal 46v, and, to a lesser extent, middle 12r. This is also reflected by the dPCA showing, in these areas, a clear separation of the trajectories relative to the different stimuli. Interestingly, when considering the Video task, trajectories relative to stimuli belonging to the same semantic category [45] appear to be grouped together within these areas. These functional properties strongly resemble those described by previous electrophysiological studies on caudal prefrontal areas [15,27,50,60] and high-order temporal visual areas [61,62] and are in line with the fact that caudal 12r, caudal 46v, and middle 12r show strong connections with temporal visual areas [20–22]. Thus, we propose that these prefrontal areas represent the first stage of processing of the visual input reaching VLPF. Considering the Visuo-Motor task, in line with the data obtained in the passive tasks, decoding analyses show that caudal areas 46v and 12r and, to a lesser extent, middle 12r present a clear pattern of object coding which is shared between the phases of the task going from object presentation onward, indicating that caudal and ventral VLPF areas are more strongly involved in coding object-related visual information also when the stimuli are actively used to guide the behavior.

Among caudal areas, 45A appears less involved in visual processing. Indeed, in this area, we recorded a smaller number of responses to passively observed visual stimuli than in caudal 12r and caudal 46v. Noteworthy, this area has strong connections with posterior temporal areas providing visual and auditory information, including communicative stimuli, and with prefrontal regions and subcortical structures involved in oculomotor control and gaze orientation [39,40,63]. Thus, it is possible that the lower number of visual responses in this area with respect to the other caudal VLPF regions is due to the fact that its neurons typically respond to acoustic stimuli or to visual-acoustic stimuli combination [64,65], which were not included in our paradigms.

## The intermediate region of VLPF exploits visual information for guiding actions

Our Visuo-Motor task allowed us to observe that the middle part of the investigated region is deeply involved in exploiting visual information for action guidance. Considering Condition coding in the Visuo-Motor task, middle 46v and, to a lesser extent, middle 12r and caudal 46v, are characterized by the strongest Condition-related differentiation (dPCA) and, as observed with the decoding analysis, by a very high accuracy in discriminating between conditions, with a stable coding

observed across the initial and final phases of the task (see below). It is worth noting that in the Visuo-Motor task the monkeys can already decide whether to act or withhold actions during the Cue period, based on cue color (green or red, respectively), but the decision on the specific motor program to employ in the Action condition can only occur after object presentation. Thus, the static pattern observed between the initial and final phases of the task could partly reflect the structure of our task.

In addition, neurons showing a preference for the Action condition during object presentation are well represented in middle areas 46v and 12r, with some differences among monkeys. This evidence is in line with the functional gradients observed in literature [15,27], showing that these prefrontal regions have a stronger involvement in coding visual stimuli when these latter are actively exploited in relation to a subsequent behavioral response. Noteworthy, these areas are also characterized by the strongest representation of neurons showing a preference for the Action condition in the final phases of the task, hence during action programming and execution, in line with their strong connections with the parieto-premotor nodes of the grasping circuit [20,22,66].

Overall, these results indicate that the intermediate/dorsal areas of VLPF are strongly involved in encoding the behavior associated to the actual context.

### VLPF areas show partially different contributions to action organization

Within the above-described general trend, we were also able to observe some peculiarities characterizing each investigated area.

**Caudal area 46v.** Concerning caudal 46v, decoding analysis shows that this area is characterized by a pattern of Condition coding that is shared between a phase encompassing the final part of the Decision and the initial part of the Behavioral response periods and the Cue/Presentation periods. A similar pattern has been described in our previous work [47] in which we proposed that this pattern indicates that the presented cue and object are already coded in terms of behavioral outputs ('pragmatic' hypothesis, see [47]). Looking at the mean population response, task-related neurons of Caudal 46v show a preference for the Action condition during cue and object appearance and an inhibition when the eyes are free to move (after reward delivery in both conditions and, though less evident, around object holding in the Action condition). Considering that this area is strongly connected with oculomotor centers [22,39], this inhibition suggests that the prospective coding observed in the decoding analysis could be linked to the oculomotor control.

**Middle area 46v.** Considering middle 46v, the decoding analysis shows a very stable representation of the Condition from the Presentation period onwards, and dPCA shows that this area is characterized by a very strong differentiation between Conditions, especially in the final phase of the task. In addition, the mean population response observed in this area revealed a clear preference for the Action condition from object presentation to the end of the task. The same pattern of activity has been described in our previous work [47], when considering the population of neurons recorded in the whole VLPF showing a preference for the Action condition. Altogether, these data indicate that neurons of middle 46v encode information related to the presented visual stimuli in terms of the subsequent goal to be achieved rather than in purely visual terms [2,62–64]. In other words, neurons of middle 46v encode objects in pragmatic terms, namely as predictions of the critical events signaling the behavioral outcome (e.g., taking possession and pulling the object and reward delivery, 'pragmatic' hypothesis; [47]).

These results are in line with electrophysiological studies showing that prefrontal neurons encoding the behavioral output during the execution of a motor task are located mainly in the intermediate portion of area 46v [34–36,67,68], as well as with its strong parietal and premotor connectivity [22,25,38]. These connections could provide middle 46v with the motor signals and sensory (somatosensory and visual) information necessary for the on-line control of behavior and for monitoring the achievement of the final action goal.

**Caudal area 12r.** In Caudal 12r, Condition decoding reveals a temporally stable pattern that emerges with object presentation and persists until the final phases of the task, closely resembling the generalization profile observed for

Object decoding. This similarity suggests that Condition-related information, in this area, is represented in a format closely tied to object-related encoding. Given the above-mentioned strong involvement of this area in processing passive visual stimuli, we propose that the coding of contextual information observed in the Visuo-Motor task reflects predominantly a visual coding. This interpretation is further supported by anatomical evidence showing that Caudal 12r has extra-prefrontal connections primarily limited to high-order inferotemporal areas [20].

**Middle area 12r.**  Concerning middle 12r, Condition decoding revealed a shared representation of this factor between Decision and Cue periods. A similar pattern was identified by Rozzi and coworkers [47] when analyzing the population of Cue-related neurons showing a preference for the Inaction condition. This suggests that cue appearance in the Inaction condition is coded in terms of the expected feedback signaling goal achievement (i.e., withholding hand movements till reward). Altogether, these findings suggest that middle area 12r may be primarily involved in exploiting visual contextual information to guide behavior. This interpretation is supported by the strong connections of this area with high-order inferotemporal regions, and with the parieto-premotor circuit involved in hand grasping actions [20,69,70].

**Area 45A.**  Area 45A is the only sector in which Cue responses are more represented than those to the other periods of the task. In addition, the mean population response is strongly enhanced during cue and object presentation and slightly inhibited during fixation maintenance (Behavioral response period of the Inaction condition). These findings are in line with the strong connections of this area with parietal and frontal areas and subcortical centers involved in oculomotion [40]

The nonspecific visual responses of neurons of this area to behaviorally relevant information could be related to possible attentional processes, in line with a series of observation indicating a specific involvement of this region in attention-related functions linked to the visual search of specific targets [71,72].

**Rostral sector.**  The rostral sector of the investigated region of VLPF is much less responsive to the tasks employed in this work. This does not appear to depend on a low numerosity of recorded neurons, since this is quite similar to that of the other investigated areas. The low responsiveness could be due to the fact that our tasks were not able to probe the specific functions of this region. This negative result, however, allow us to define a clear border between the above-described areas and this rostral territory.

The time course of the population response of this area is characterized, in the Visuo-Motor task, by an interesting functional feature: the activity is suppressed during the whole task unfolding, in both Action and Inaction conditions, except for two excitatory peaks in correspondence of stimuli presentation. This inhibition could be interpreted in light of a large corpus of literature showing that the rostral prefrontal cortex is deeply involved in several operations, such as coding behavioral rules based on past episodic information, coding self-generated decisions, and storing conscious action plans [11–13,73,74]. As the task strongly depends on a clearly defined and immediately available context and is well learned by monkeys, the VLPF areas active during both Action and Inaction conditions could in the meantime inhibit the activity of rostral VLPF. This explanation is corroborated by the known strong intra-prefrontal connectivity of the rostral VLPF sector with more posterior VLPF regions [20,22].

## Limitations and further developments

Usually, neurophysiological studies test functions of cortical regions by using a task specifically aimed to evaluate the effect of few variables and by limiting the analysis to task-related neurons. In this paper, we employed three different tasks to better evaluate multiple functional variables on the same neurons. Furthermore, although in the main text we present the properties of task-related neurons, we also demonstrate that the results are confirmed, for each area, in the whole population of recorded neurons (see Supporting information). Thus, we can infer that the properties of the population of task-related neurons are representative of those of the whole investigated region within the studied functions.

However, the present work has, as a main limitation, the fact that the employed tasks only partially allowed us to characterize the functions of 45A and of the rostral sector of VLPF. Further studies could allow to better define the properties of these two cortical sectors, by verifying their response to other types of sensory stimuli (e.g., acoustic, somatosensory,

or multimodal), by assessing their role in high-level guidance of oculomotor behavior, and in the processing of abstract information stored in episodic memory.

The types of motor behavior studied in this work, i.e., grasping objects and withholding actions, are daily performed also by humans, therefore, our results could be important for verifying whether a modular organization of cognitive-motor functions, similar to that here described in the monkey VLPF, is present also in our species. Interestingly, in many prefrontal neuropsychological syndromes, such as utilization behavior, echopraxia, and anarchic hand syndrome, the impairment of cognitive aspects is intrinsically linked to the motor ones [75–77]. Thus, we believe that the detailed anatomical identification of the prefrontal areas involved in different aspects of executive functions, such as context-based organization of behavior, will allow a more precise "mapping" of the symptoms of prefrontal syndromes.

## Materials and Methods

### Subjects and ethical approvals

The experiment was carried out on two female Rhesus monkeys (*Macaca mulatta*, M1, M2) weighing about 4 kg. The animals have been previously employed in a series of experiments, whose results have already been published [36,45–47]. All methods were carried out in accordance with the European (2010/63/EU) and the ARRIVE guidelines. The experimental protocols, the animal handling, and the surgical and experimental procedures complied with the European guidelines (2010/63/EU) and Italian laws in force on the care and use of laboratory animals, and were approved by the Veterinarian Animal Care and Use Committee of the University of Parma (Prot. 78/12 17/07/2012) and authorized by the Italian Ministry of Health (D.M. 294/2012-C, 11/12/2012).

### Training and surgical procedures

The monkeys were first habituated to seat on a primate chair and to familiarize with the experimental setup. At the end of the habituation sessions, a head fixation system (Crist Instruments Co) was implanted. Then, the monkeys were trained to perform the tasks described below. After completion of the training, a recording chamber (32 × 18 mm, Alpha Omega, Nazareth, Israel) was implanted on the VLPF, based on MRI scan. All surgeries were carried out under general anesthesia (ketamine hydrochloride, 5 mg/kg, i.m. and medetomidine hydrochloride, 0.1 mg/kg, i.m.), followed by postsurgical pain medication.

### Experimental apparatus

During training and recording sessions, the monkeys seated on the monkey chair with the hand contralateral to the hemisphere to be recorded on a resting position, located 9 cm in front of the abdomen. A monitor was positioned in front of the monkey, to present the visual stimuli used in the passive tasks (Picture and Video tasks, see below). The monitor, with a resolution of 1680×1050 pixel, was positioned at 54 cm from the monkey's face, and its geometrical center was located at the height of monkey's eyes. A laser spot could be projected on the center of the screen as a fixation point. A phototransistor was placed on the monitor in order to provide the onset and offset of the visual stimuli.

During the Visuo-Motor task, a box containing three objects was positioned at 22 cm from the monkey's chest. The opening of a small door (7 × 7 cm) in the frontal panel of the box at the height of monkey's eyes allowed to present the three objects, one at the time. Two laser spots (instructing cues) of different colors (green and red) could be projected onto the box door or onto the object, signaling the task conditions and phases.

### Behavioral paradigms and stimuli

**Picture task.** The Picture task corresponds to the *Visual task* described in [46]. Briefly, to evaluate the response of VLPF neurons to the observation of static visual stimuli, 12 different images (6° × 6°; see below) were presented, while

the monkeys kept their gaze within a 6° × 6° fixation window centered on the stimulus. Fig 1A shows the sequence of events occurring during each trial. The monkeys were required to keep their hand on the resting position; if this was accomplished, the trial started, and the fixation point (red laser spot) was turned on, and they had to fixate it for a randomized time interval (500–900 ms). If they kept fixation for this period of time, the fixation point turned off and one of the images was presented for 600 ms, centered on the fixation point. The monkeys had to observe it (without breaking fixation) throughout the presentation period. Then, the image disappeared, the fixation point turned on again for a randomized period (500–900 ms) and the monkeys had to keep fixation on it.

The trials were accepted as correct, and the monkeys were rewarded, if they kept their eyes within the fixation window for the duration of each phase of the task and did not release the hand from the resting position. Discarded trials were repeated at the end of the sequence to collect at least 10 presentations for each stimulus. The order of stimuli presentation was randomized.

The 12 stimuli (Fig 1A) belong to 4 different semantic categories:

- Graspable solids (pictures of the objects employed in the motor task described in [36,47]: cube, cylinder, sphere);

- Fruits: apple, banana, peanut;

- Faces: human face, monkey face, sketchy drawing of a face;

- Laboratory furniture, geometric, but not graspable: shelf, monitor and clock.

The stimuli were homogeneous for luminance. Note that the fruits and solids could evoke similar affordances (apple and cube: power grip; banana and cylinder: finger prehension; sphere and peanut: precision grip).

**Video Task.** The Video task corresponds to that described in [45]. In order to evaluate the response of VLPF neurons to observation of dynamic visual stimuli, we displayed videos (12° × 12°) showing several biological stimuli and object motion (see below), while the monkey maintained fixation within a 6° × 6° fixation window centered on the video. The sequence of events occurring during each trial is the same as in the Picture task (see Fig 1B), but the stimuli were presented for 1800 ms. The criteria employed for trial acceptance were the same as in the Picture task (see above).

Discarded trials were repeated at the end of the sequence in order to collect at least 10 presentations for each stimulus. The order of stimuli presentation was randomized.

The construction of the set of video stimuli was devised so to present to the monkey goal-related or non-goal-related actions, different agents and presence or absence of an object. Specifically, the six stimuli were the following (Fig 1B):

- Monkey grasping in first-person perspective (*MGI*): a monkey right forelimb enters into the scene from the lower border of the video, reaches and grasps a piece of food located in the center of it, and lift it toward itself (only the initial phase of this latter movement is visible). The observed forelimb is presented as if the observing monkey was looking at its own forelimb during grasping.

- Monkey grasping in third person perspective (*MGIII)*: a monkey, located in front of the observer, with its left forelimb reaches and grasps a piece of food located in the center of the video, and brings it toward itself.

- Human grasping (HG): a human actor, located on the right of the video reaches, grasps and lifts an object, located in the center of the video, with his right forelimb.

- Human mimicking (HM): a human actor, located on the right of the video, performs the pantomime of the same action shown in HG, without the object.

- Biological movement (BM): a human actor located on the right of the video extends his right forelimb, with the hand open, to reach the central part of the screen. No object is present.

- Object motion (OM): an object is presented in the center of the screen and moves along the same trajectory as in HG. This stimulus was obtained by removing the agent from HG, in order to have same stimulus kinematics as in HG.

In the videos, the agents' faces were not shown in order to avoid the possible influence of neural responses due to face presentation.

**Visuo-Motor task.** The Visuo-Motor task is the same described in [36,47]. Briefly, the task consisted of two basic conditions: Action and Inaction (Fig 1C). Each trial started with the monkeys' hand on the starting position. Then, one of the two instructing cues (green = Action condition; red = Inaction condition) was turned on and projected onto a closed box door, placed in front of the monkeys. In both conditions, the monkeys had to maintain fixation within a 6° × 6° fixation window centered on the instructing cue for a randomized time interval (500–1,100 ms). Then, the box door opened allowing the monkeys to see one of three objects.

In the Action condition, during object presentation, the monkeys had to maintain fixation with the green cue still on, projected onto the object. After a randomized time (700–1,100 ms), the green cue turned off (Go signal), instructing the monkeys to reach for, grasp the object and pull it.

In the Inaction condition, the trial unfolding and the events' timing were the same as in the Action condition till the red cue turned off, after which the monkeys were required to keep fixation for a further 600 ms period, refraining from acting. The order of presentation of both objects and conditions was randomized.

If the monkeys correctly performed a trial, the reward was delivered at the end of it. A trial was discarded when one of the following types of error occurred: (1) releasing the hand from the resting position before reward delivery in the Inaction condition or before the Go signal in the Action condition; (2) breaking fixation before reward delivery in the Inaction condition or before the Go signal in the Action condition; (3) failing to reach and grasp the object; (4) grasping the object with an incorrect prehension. Discarded trials were repeated at the end of the sequence to collect at least 30 correct trials for condition (10 trials × 3 objects).

### Recording techniques, task events acquisition and microstimulation

Neuronal recordings were performed by means of a multi-electrode recording system (AlphaLab Pro, Alpha Omega Engineering, Nazareth, Israel) employing a maximum of eight glass-coated microelectrodes (impedance, 0.5–1 MΩ) inserted through the intact dura. The microelectrodes were mounted on an electrode holder (MT, Microdriving Terminal, Alpha Omega) allowing electrodes displacement, controlled by a dedicated software (EPS; Alpha Omega). The MT holder was directly mounted on the recording chamber. Neuronal activity was filtered, amplified, and monitored with a multichannel processor and sorted using a multi-spike detector (MCP Plus 8 and ASD, Alpha Omega Engineering). Spike sorting was performed using the Off-line Sorter (Plexon, Dallas TX, USA). During each recording session, electrodes were inserted one after the other inside the dura until the first neuronal activity was detected for each of them. Each electrode was then deepened into the cortex independently one from the other, in steps of 500 μm until the depth at which the border between gray and white matter was reached (see [78] for a similar procedure). At each site, multiunit and single-unit activities were recorded for subsequent analyses. In the present study, we considered the activity recorded at three levels of depth (500, 1,000, 1,500 μm). The experiment was controlled by a homemade Labview software. Digital output signals determined the onset and offset of laser spots, image/videoclip presentation, opening of the door and reward release. Contact-detecting electric circuits provided the digital signals related to monkey hand contact and release of the resting position and the beginning and end of object pulling.

In order to identify the sector where eye movements can be elicited by intracortical microstimulation, the recording microelectrodes were also used for delivering intracortical monophasic trains of cathodic square wave pulses, through a constant current isolator (World Precision Instruments, Stevenage, UK) with the following parameters: total train duration, 50 ms or, when no response was elicited, 100 ms; single pulse width, 0.2 ms; pulse frequency, 330 Hz. The stimulation

started with a current intensity of 100 μA that was decreased until threshold definition and was controlled on an oscilloscope by measuring the voltage drop across a 10 KΩ resistor in series with the stimulating electrode.

Eye movements were recorded using an infrared pupil/corneal reflection tracking system (Iscan, Cambridge, MA, USA) positioned above the box. Sampling rate was 120 Hz.

### Histology, reconstruction of the recorded area, and identification of the regions of interest on the basis of architectural and connectional data

Before sacrificing the animals, electrolytic lesions (10 μA cathodic pulses per 10 s) were performed at known coordinates at the external borders of the recorded region. After 1 week, each animal was anaesthetized with ketamine chloride (15 mg⁄kg i.m.), followed by an i.v. lethal injection of pentobarbital sodium and perfused through the left cardiac ventricle with buffered saline, followed by fixative. The brain was then removed from the skull, photographed, frozen, and cut coronally. Each second and fifth section (60 μm thick) of a series of five were stained using the Nissl method. The locations of penetrations were then reconstructed on the basis of electrolytic lesions, stereotaxic coordinates, depths of penetrations, and functional properties. More specifically, penetrations deeper than 3,000 μm located inside the Arcuate sulcus and Principal Sulcus were used in order to localize the posterior and dorsal border of VLPF, respectively.

In order to define anatomical areas in the recorded brains, we performed a parcellation relying on previous architectonical and connectional studies from our lab [20–22,40]. In those studies, we first defined the architectonic borders of areas 8FEF, 8r, 45A, 45B, 46v, and 12r and, subsequently, we injected neural tracers in these areas on 12 monkeys (see S1 Table); the different pattern of connections observed after each injection confirmed the architectonic borders and allowed to further subdivide areas 46v and 12r in three additional sectors. In the present work, we superimposed the areas and sectors identified in the above-mentioned studies onto the histological reconstructions of the two recorded brains, by warping both the architectonic maps and the locations of the various injection sites used to characterize the connectivity of each area. Specifically, this warping was performed using a non-linear transformation procedure (for details see [48]) based on specific anatomical anchor points: the anterior and posterior tips of the principal sulcus, the spur of the arcuate sulcus, the tips of the superior and inferior limbs of the arcuate sulcus, the tips of the superior and inferior prefrontal dimples, and the orbital reflection at the level of the lateral orbital sulcus.

In addition, we defined an oculomotor prearcuate sector by using microstimulation and recording of saccade-related neuronal activity. More in details, concerning microstimulation, oculomotor sites were those in which saccadic eye movements could be elicited with currents lower of or equal to 60 μA, with a train duration of 50 or 100 ms; concerning single neuron recordings, we described saccade-related activity by aligning the neuronal discharge with the moment in which the eyes reached a fixation target. This region, overlapping the location of areas 8 and 45B, has been excluded from further analyses.

The location of our recording chamber allowed us to record only from a small number of sites falling in areas rostral 46v and rostral 12r, especially in M2, thus, we pulled together the data obtained from these areas and named this region *rostral sector*.

### Analysis of single neurons responses

The digital signals representing the different task events, described above, were employed to align neuronal activity and to create the response histograms and data files used for the statistical analyses described in the subsequent paragraphs.

**Picture task.** We recorded neuronal activity for at least 120 successful trials, 10 for each stimulus. For the statistical analysis, two epochs were defined (see [46]): (1) *Baseline*: 500 ms preceding stimulus presentation, during which the monkey was looking at the fixation point; (2) *Presentation*: the first 500 ms of image presentation.

Single-neuron responses were statistically evaluated by means of a 2×12 ANOVA (Factors: Epoch, Stimulus, $p < 0.01$) followed by Newman-Keuls post-hoc tests.

A neuron was considered as task-related when the 2×12 ANOVA revealed: (1) a significant main effect of the Epoch factor ($p < 0.01$) and/or (2) a significant interaction effect (Epoch × Stimulus, $p < 0.01$), with the post-hoc tests showing a significant difference between at least one *Presentation* epoch of one image and its corresponding *Baseline epoch*. Task-related neurons were classified as *selective* when the 2×12 ANOVA revealed a significant Interaction effect and the post-hoc test showed a significant difference among the activity recorded in the *Presentation epoch* of one image and that of its corresponding *Baseline epoch* as well as a significant difference between the activity recorded in the *Presentation epoch* of that image and the *Presentation epoch* of at least another image. Neurons were classified as *unselective* when the statistical test revealed a significant Main effect of the Epoch factor and/or a significant Interaction effect, and the post-hoc test did not show any difference among the activities recorded in the *Presentation* epochs of the 12 images.

**Video task.** We recorded neuronal activity for at least 60 successful trials. For the statistical analysis, three epochs were defined (see [45]): (1) *Baseline*: 500 ms before the beginning of the videos, during which the monkey was looking at the fixation point; (2) *Video Epoch 1*: the first 700 ms of the videos (except for MGI, where, because of the fastest arm movement, the epoch lasted 500 ms); (3) *Video Epoch 2*: the subsequent 700 ms of the videos. Note that Epoch 1 includes the context of the scene and the beginning of the forelimb movement, while Epoch 2 includes the hand-object contact in the case of action and the end of movement/object motion in all other videos.

Single-neuron responses were statistically evaluated by means of a 3×6 ANOVA (Factors: Epoch, Stimulus, $p < 0.01$) followed by Newman–Keuls post-hoc tests.

A neuron was considered as task-related when the 3×6 ANOVA revealed: (1) a significant main effect of the Epoch factor ($p < 0.01$), with post-hoc tests indicating a significant difference between at least one of the two *video epochs* and the baseline and/or (2) a significant interaction effect (Epoch × Stimulus, $p < 0.01$), with post-hoc tests showing a significant difference between at least one of the *video epochs* of one video and the corresponding *baseline* epoch.

Task-related neurons were classified as *selective* when the 3×6 ANOVA revealed a significant Interaction effect and the post-hoc test showed a significant difference among the activity recorded in one of the two *Video epoch*s of one video and that of the corresponding *Baseline epoch* as well as a significant difference between the activity recorded in one of the two *Video epochs* and that recorded in the same epochs of at least another video. Neurons were classified as *unselective* when the statistical test revealed a significant Main effect of the Epoch factor and/or a significant Interaction effect, and the post-hoc test did not show any difference among the activities recorded in the *Video epochs* of the 6 videos.

**Visuo-Motor task.** We recorded neuronal activity for at least 60 successful trials (30 per condition, 10 for each object). For statistical analysis of the neural activity, nine epochs have been defined (see [36,46]), based on the digital signals: (1) Baseline: from 750 to 250 ms before the onset of the instructing cue; (2) Pre-cue: 250 ms preceding the onset of the instructing cue; (3) Cue: 250 ms following the onset of the instructing cue; (4) Pre-presentation: 500 ms preceding the opening of the box door; (5) Presentation: 500 ms following door opening (object presentation); (6) Set: 250 ms before the offset of the instructing cue; (7) Go/NoGo, from the offset of the instructing cue to the release of the hand starting position (Action condition) or 250 ms following the offset of the instructing cue (Inaction condition); (8) Grasping/Fixation: from 250 ms before to 250 ms after the Pulling onset (Action condition) or a time period ranging from 250 to 500 ms after the offset of the instructing cue (Inaction condition) Reward: 500 ms following reward delivery.

Single-neuron responses were statistically evaluated by means of a 9×2 ANOVA (Factors: Epoch, Condition, $p < 0.01$) followed by Newman-Keuls post-hoc tests. Since trials were randomized, changes of the baseline activity across trials were not expected, and the neurons showing a significant difference between baselines were discarded. Neurons were included in our dataset and were defined as *task related* when the 9×2 ANOVA revealed at least one of the two following effects: 1) a significant main effect of the Epoch factor ($p < 0.01$), with the relative post-hoc tests showing a significant difference between the activity recorded in the Baseline epoch and in at least one of the other epochs

(Condition-independent neurons); (2) a significant Interaction effect (Condition × Epoch, $p < 0.01$), with the subsequent post-hoc tests showing a significant difference between at least one epoch of one condition and both its baseline and the corresponding epoch of the other condition (Condition-dependent neurons). Considering that the epochs of Pre-cue and Reward fall in the inter-trial period, when eye movements are not controlled, we decided to consider, for our analysis, the remaining six epochs plus the Baseline.

## Data matrix construction for analyses at the population level

After single neuron analysis, we performed further analyses relying on the same data matrix and time periods. Since in all tasks most time intervals were variable (see above), to perform these analyses, we aligned neural activity on different task events, segmenting it in 20 ms bins, and considered specific time periods for each task, locked on these events, as follows.

**Picture and Video tasks.** Baseline: 500 ms preceding fixation point onset; Presentation: a period of 500 ms (picture task) or 1,400 ms (video task) following stimulus appearance.

**Visuo-Motor task.**

Baseline: from 750 to 250 ms before the onset of the instructing cue;

Cue: 500 ms following cue onset;

Presentation: 500 ms following object presentation;

Decision: 400 ms centered on the Go/NoGo signal;

Behavioral response: a 400 ms period, starting, in the Action condition, 300 ms before object holding, and in the Inaction condition, 200 ms after the NoGo signal.

## Unsupervised clustering of neuronal responses

To identify functional clusters of neurons and assess their anatomical distribution, we performed an unsupervised clustering of the neuronal responses observed in the three behavioral paradigms. Starting from each data matrix, we calculated, for each 20 ms bin, the mean firing rate over trials belonging to each combination of the two conditions and three objects in the Visuo-Motor task, or to the same type of stimulus in the Picture or Video Task. The mean firing rate of the baseline period was then subtracted from the firing rate of each bin, obtaining the final matrices on which the cluster analyses were performed. We then applied a non-linear dimensionality reduction method, UMAP [79], to obtain a simplified representation of these matrices. Subsequently, we performed a K-means clustering on these data, using the MATLAB *evalclusters* function, which allowed us to subdivide the considered neurons into different clusters. The optimal number of clusters was selected by using the "cluster silhouette" metric, ranging from 1 to 10 possible clusters. Finally, to map the spatial distribution of the neurons, we calculated, for each recording site, the percentage of neurons belonging to each identified cluster, out of the total number of task-related neurons recorded on that site (three groups: 1%–30%, 31%–60%; 61%–100%). Then, we superimposed the resulting map onto the 2D anatomical reconstruction of the recorded brains. To more effectively show the spatial distribution of clusters, we outlined the regions hosting at least two adjacent sites. Isolated sites were considered only if containing at least 31% of neurons belonging to the cluster.

## Time course of mean population activity

To characterize the time course and the discharge rate of the considered neuronal populations with respect to the main tasks phases, the neuronal activity of each population was aligned with the main behavioral events. The population activity was computed as follows. The mean single neuron activity over trials, in terms of firing rate, was calculated for each 20

ms bin in the two conditions. The average baseline activity was then subtracted from the mean single neuron activity over trials for each bin. Thus, in this analysis, 0 represents baseline activity. No normalization on the maximal firing rate was performed. Each neuron contributed one entry to each data set.

## Demixed principal component analysis

In order to evaluate how the population of neurons recorded in each area encodes specific factors during the unfolding of the three tasks, we adopted a data-simplification method: the demixed principal component analysis (dPCA), using freely available code provided by Kobak and coworkers ([80], see also [47]).

We performed this analysis starting, for each task, from the data matrix described above, and considering the factor 'Type of Stimulus' for the Picture and Video tasks (including the 12 stimuli and 6 videos, respectively), and the factors 'Condition' (Action and Inaction) and 'Object' (Cube, Cylinder and Sphere) for the Visuo-Motor task. This analysis allowed us to extract demixed principal components that captured task-related variance, either specific to the individual factors or shared across them (Factor-independent). Specifically, we considered the demixed principal components explaining the largest proportions of *Condition*, *Object*, and *Factor-Independent* variance in the Visuo-Motor task, and of *Type of Stimulus* and *Factor-Independent* variance in the Picture and Video tasks. We used the selected components to define low-dimensional subspaces that isolate the neural dynamics associated with each experimental factor These dynamics were represented as trajectories obtained by projecting the baseline-corrected mean population activity onto the corresponding subspaces (see [49] for a similar approach).

## Decoding analysis

In order to evaluate the temporal evolution of information coding by the different neuronal populations recorded in the Visuo-Motor task and if this information is encoded in static or dynamic patterns of activity, we adopted a population decoding approach according to the methodology described by Meyers and coworkers [81,82]. Decoding analysis was performed as described in [47], using the data matrix described above. In particular, for each neuron, trial-by-trial average firing rates were computed in 60 ms bins, sampled every 20 ms. Each obtained data point was then labeled according to the decoding factor of interest (Condition or Object). Data points were then randomly grouped into $k$ non-overlapping splits, where $k$ matched the number of data points per class (30 for Condition decoding, 20 for Object decoding). Each split included a pseudopopulation of neurons, i.e., neurons recorded separately but treated as if recorded simultaneously. Next, a Poisson naïve Bayes classifier was trained on $k − 1$ splits and tested on the held-out split, in a $k$-fold cross-validation scheme. Finally, this entire procedure was repeated 50 times with different random splits, and decoding accuracy was averaged across repetitions to improve robustness. Note that, with respect to the results of cross-temporal decoding, our definition of *static patterns of coding* refers to the situations in which the accuracy of an off-diagonal bin is significantly greater than chance and does not differ statistically from the accuracy values of the respective on-diagonal bins (see the next section and [82]). The data alignment on task events and the binning procedure described above led to merge in the same bin the activity at the border between two subsequent periods of the task (bins of 60 ms, sampled at 20 ms intervals). Accordingly, in our analysis, we removed the last two bins of each task period considered in the analysis, obtaining a total number of 105 considered time bins.

**Classification of static patterns.** To investigate whether the observed *static patterns of coding* are statistically significant, we used a method employed in previous studies [82,83]. First, for each off-diagonal bin, we calculated the differences between its accuracy value and that of the two on-diagonal bins used for training and testing. The same procedure was repeated 1,000 times using accuracies obtained after shuffling the labels, allowing us to obtain a null distribution of the differences in accuracy values between the off- and on-diagonal bins. For statistical analysis, we selected the "significant" time bins, defined as those in which both differences were lower than 99.9% of the differences

estimated at the corresponding time points in the null distribution. Then, we performed a cluster-based test [84], by comparing the summed difference values of the observed-data clusters with the maximum summed cluster values of the null distribution. In particular, we identified clusters of consecutive "significant" off-diagonal time bins (n. bins > 0), and summed the differences observed in each point of the cluster. The same procedure was applied using the accuracy value matrices resulting from the 1,000 shuffled decodings, each time extracting the maximum summed cluster values. The proportion (over the 1,000 iterations) of times in which the maximum summed cluster values of the null distribution were lower than the values obtained in each observed-data cluster determined the $p$-value of the test. The bins for which this $p$-value was below 0.001 were considered significantly static off-diagonal bins. Similar to [82], as a further restrictive criterion to classify the off-diagonal bins as significantly static, the two corresponding on-diagonal bins used for training and testing the classifier were required to be both significantly above chance level. In this case, we performed a permutation test consisting in evaluating the proportion of times in which the accuracy value observed in each on-diagonal time bin was higher than that observed in the null distribution, which determined the $p$-value of the test. The accuracy in an on-diagonal time bin was then considered significantly above chance if the obtained $p$-value was below 0.000009, corresponding to a $p$-value of 0.001 Bonferroni corrected for the 105 considered bins.

## Supporting information

**S1 Table. Injection sites in the VLPF areas: monkey species, localization of the injections and type and amount of injected tracers.**
(PDF)

**S1 Fig. Spatial distribution and response of task-related neurons belonging to different clusters.** Distribution of task-related neurons belonging to each identified cluster in the Picture, Video, and Visuo-Motor tasks. The distribution of each cluster is represented by colored outlines covering the regions hosting at least two adjacent sites or hosting isolated sites containing more than 31% of neurons belonging to the cluster (see Materials and methods). For each monkey, the temporal profile of the mean baseline-subtracted activity of neurons belonging to each identified cluster is represented on the right of the respective map. The dashed line indicates baseline-level activity. The neuronal activity is aligned on the main task events indicated below each panel of the figure (see Materials and methods for details on the considered periods).
(EPS)

**S2 Fig. Spatial distribution of selective neurons belonging to different clusters.** Distribution of selective neurons belonging to each cluster identified by considering the Presentation periods of the Picture and Video tasks, the final phase of the Visuo-Motor task, and the Cue and Presentation periods of the Visuo-Motor task. Blue and violet outlines represent the distribution of clusters identified analyzing the Picture and Video task, respectively. Green outlines represent the distribution of clusters identified analyzing the final phases of the Visuo-Motor task. Yellow and light blue outlines represent the distribution of clusters identified analyzing the Cue and Presentation periods of the Visuo-Motor task. Other conventions as in S1 Fig.
(EPS)

**S3 Fig. dPCA trajectories of all recorded neurons of each VLPF area in the Picture task.** Each panel depicts the time course of the first Factor-Independent ($X$ axis) and of the first Type of Stimulus-related ($Y$ axis) principal components plotted together, relative to the population of all neurons recorded in each area. Each colored line corresponds to one of the 12 stimuli presented in the task. Green squares represent the time at which the stimulus presentation occurs. The data matrices (see Materials and methods) underlying this figure can be found in OFS database: https://osf.io/j8fcs/.
(EPS)

**S4 Fig. dPCA trajectories of all recorded neurons of each VLPF area in the Video task.** Each panel depicts the time course of the first Factor-Independent (*X* axis) and of the first Type of Stimulus-related (*Y* axis) principal components plotted together, relative to the population of all neurons recorded in each area. Each colored line corresponds to one of the six videos presented in the task. Green squares represent the time at which the stimulus presentation occurs. The data matrices (see Materials and methods) underlying this figure can be found in OFS database: https://osf.io/j8fcs/.
(EPS)

**S5 Fig. dPCA trajectories of all recorded neurons of each VLPF area in the Visuo-Motor task.** Each panel depicts the time course of the first Condition-related (*X* axis), of the first Object-related (*Y* axis) and of the first Factor-Independent (*Z* axis) principal components, relative to the population of all the neurons recorded in each area. Green and Magenta colored lines represent Action-related and Inaction-related trajectories, respectively. Continuous and dashed lines correspond to the three objects presented, respectively. Blue circles, brown squares, and orange triangles represent, for each trajectory, the time of cue onset, object presentation, and cue offset, respectively. The data matrices (see Materials and methods) underlying this figure can be found in OFS database: https://osf.io/j8fcs/.
(EPS)

**S6 Fig. Mean activity of the whole population of neurons recorded in each area during the Visuo-Motor task.** Temporal profile of the mean net activity of the whole population of recorded neurons (including both task-related and non-task-related neurons). Magenta and green curves indicate the population mean net activity in the Inaction and Action condition, respectively. Other conventions as in Fig 8. The data matrices (see Materials and methods) underlying this figure can be found in OFS database: https://osf.io/j8fcs/.
(EPS)

**S7 Fig. Cross-temporal decoding of the Condition (A–F) and Object (A′–F′) factors of the Visuo-Motor task in the whole population of recorded neurons.** For each analysis, the decoding accuracy is computed in bins of 60 ms, sampled at 20 ms intervals. For each plot, the vertical and horizontal lines delimit the considered time periods (see Materials and methods). Other conventions as in Fig 9. The data matrices (see Materials and methods) underlying this figure can be found in OFS database: https://osf.io/j8fcs/.
(EPS)

## Acknowledgments

We thank M. Bimbi for his technical help in setting up the experiment.

## Author contributions

**Conceptualization:** Leonardo Fogassi, Stefano Rozzi.

**Formal analysis:** Claudio Basile, Marzio Gerbella, Alfonso Gravante, Amelia Lapadula, Luciano Simone, Stefano Rozzi.

**Funding acquisition:** Leonardo Fogassi.

**Investigation:** Francesca Rodà, Luciano Simone, Stefano Rozzi.

**Methodology:** Leonardo Fogassi, Stefano Rozzi.

**Visualization:** Claudio Basile, Marzio Gerbella, Amelia Lapadula.

**Writing – original draft:** Claudio Basile, Marzio Gerbella, Amelia Lapadula, Leonardo Fogassi, Stefano Rozzi.

**Writing – review & editing:** Alfonso Gravante, Francesca Rodà, Luciano Simone.

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
