## [Editor Report · Decision Letter 0]

27 Jan 2025

***HAVE YOU ASSIGNED THE AE? IF NOT CANCEL THIS DECISION AND GO BACK!***

***REMEMBER TO CLOSE ALL REDUNDANT AE DISCUSSIONS***

Dear Dr Rozzi, 

Thank you for submitting your manuscript entitled "Encoding of visual stimuli and behavioral goals in distinct anatomical areas of monkey ventrolateral prefrontal cortex" for consideration as a Research Article by PLOS Biology.

Your manuscript has now been evaluated by the PLOS Biology editorial staff [as well as by an academic editor with relevant expertise *EDIT AS APPLICABLE*] and I am writing to let you know that we would like to send your submission out for external peer review.

Once your full submission is complete, your paper will undergo a series of checks in preparation for peer review. After your manuscript has passed the checks it will be sent out for review. To provide the metadata for your submission, please Login to Editorial Manager (https://www.editorialmanager.com/pbiology) within two working days, i.e. by Jan 29 2025 11:59PM.

Kind regards,

Taylor

Taylor Hart, PhD, 

Associate Editor

PLOS Biology

thart@plos.org

---

## [Decision Letter · Decision Letter 1]

2 Apr 2025

Dear Dr Rozzi,

Thank you for your patience while your manuscript "Encoding of visual stimuli and behavioral goals in distinct anatomical areas of monkey ventrolateral prefrontal cortex" was peer-reviewed at PLOS Biology. It has now been evaluated by the PLOS Biology editors, an Academic Editor with relevant expertise, and by several independent reviewers. 

In light of the reviews, which you will find at the end of this email, we would like to invite you to revise the work to thoroughly address the reviewers' reports.

As you will see, the reviewers generally find the study to be solid, and describe the reported data as rich. However, R2, R3, and R4 raised concerns about some aspects of the analyses, as well as missing technical details. R3 and R4 suggested areas where the analyses could be expanded to increase the strength of the reported findings, and the text could be re-organized to improve comprehension.

We would like to invite you to perform a Major Revision of your manuscript. You should carefully consider the points raised by each reviewer and thoroughly address them. Based on our discussion with the Academic Editor, we would like to emphasize the need to make it easier for readers to extract the most relevant findings, for example by ensuring that each results section starts with a statement of the aims and ends with a clearer statement of the importance of the result. We encourage you to obtain feedback from your colleagues, or alternatively from professional writing services, in order to address the level of textual clarity.

Given the extent of revision needed, we cannot make a decision about publication until we have seen the revised manuscript and your response to the reviewers' comments. Your revised manuscript is likely to be sent for further evaluation by all or a subset of the reviewers.

**IMPORTANT - SUBMITTING YOUR REVISION**

*Re-submission Checklist*

*Published Peer Review*

*PLOS Data Policy*

*Blot and Gel Data Policy*

Sincerely,

Taylor

Taylor Hart, PhD, 

Associate Editor

PLOS Biology

thart@plos.org

REVIEWS:

Reviewer #1: Basile and colleagues recorded neural activity from a broad region of the ventrolateral prefrontal cortex (VLPF) during the execution of two passive observation tasks involving visual static and video stimuli of different categories, as well as an active, rule-based task in which visual cues instructed the monkeys to either inhibit or execute a grasping action directed at objects of various shapes. In this study, the authors subdivide the VLPF based on its cytoarchitectural and connectional characteristics. They then analyze and compare the coding properties of task-related information, considering visual stimuli, action versus inaction conditions, and graspable objects. Such partitioning of VLPF allowed the authors to identify distinct contributions of its subregions in coding the investigated information, which they claim to follow a rostro-caudal gradient consistent with those previously proposed for other cognitive processes.

The manuscript is built on top of the lab's previous expertise regarding the anatomical characterization of the frontal cortex and the paper's topic. The analyses are sound, combining classical and modern approaches at single cell and population levels, which collectively support the conclusions drawn by the authors. The statistical significances appropriately take into account the effect of multiple comparisons over time when that occurs (decoding analysis). The manuscript is well-written and clear, and the results are original and well-contextualized in the previous literature. The study limitations are clearly stated and transparently addressed. Finally, the involvement and specialization of prefrontal cortex components during the goal-directed behavior conveyed by the active task is of important interest. Therefore, for these reasons, I recommend the publication of the manuscript.

I do not have any major comments. But only some minor comments (listed in order of presentation in the manuscript). 

-Line 178-180 (Figure legend 2): It is not clear to me what "above" and "below" mean. Shouldn't they refer to the bar graphs shown and thus be "left" and "right"? 

-Line 182 (Figure legend 2): Red squares.

-Line 232: Typo, remove "for".

-Line 265-266: Typo, instead of "Figure 2K," it should be Figure 4K.

-Line 419-421: If I have not misunderstood, the use of "specific activity pattern" might be confusing for the reader, since (as also stated immediately after) cross-temporal decoding in S11B and S12B shows stability between epochs of presentation, decision, and behavioral response, suggesting a shared activity pattern between epochs.

-Line 757: It is written "video," but it should be "image".

-Line 835-842: Although this information may have been reported in previous publications using this dataset, I suggest including in this manuscript the number of electrodes used (or an average estimate) to obtain the neuronal signal in the recorded sessions. In order for the reader to have a general understanding of how many neurons could be recorded simultaneously.

-Line 1042: Although it is stated that decoding was implemented as described in previous publications using this dataset, I would also include some key details in this manuscript for clarity. Specifically, it would be helpful to specify the type of classifier used, the method for cross-validating the data and the percentage/number of test trials used, any data normalization procedures applied, and whether pseudo-population techniques were employed to account for non-simultaneously recorded neuronal populations.

-Line 1045-1047: This definition may lead to confusion and potentially conflict with the classification of static time points derived from the analysis of the cross-temporal decoding matrices. As described in the Methods section, an off-diagonal time point is classified as static if the difference in accuracy from the on-diagonal time points used for training and testing is not statistically significant. For this assessment, it is only considered that the accuracy of the off-diagonal time point is significantly greater than chance without considering its absolute magnitude value.

Line 1061-1065: Part of the method described here involves the first steps of the statistical calculation performed by the cluster-based permutation test. I would move this description a little later when the statistical test is explained in detail.

-Fig 4c: In this figure, unlike the others, the number of neurons is reported instead of percentages. Unless there is a specific reason, I would also use percentages here as well to maintain consistency across the figures.

-Fig 4n: There is a typo on the y-axis labels.

-Figure legend 4: A description of Figure 4C is missing.

-Figure S10-11: Considering that the variable "condition" and "object" lead to two different chance levels in the decoding analysis (50% and 33% accuracy, respectively), I would suggest adding a horizontal line in the plots regarding "Significant on-diagonal time bins" to make explicit for the reader the expected chance level.

Reviewer #2: Basile et al. present a study including two monkeys performing three different tasks (two passive visual tasks, one visuo-motor task) with the aim to understand whether distinct anatomical sectors of the ventrolateral prefrontal cortex play different functional roles. Their results show that distinct areas in the VLPFC contribute to visual processing and action organization along the caudo-rostral axis.

The strength of the study comes from their ability to identify distinct anatomical areas across the two monkeys based on a previous anatomical parcellation, their ability to nicely condense a rich amount of results and that they were able to have the two monkeys preform three different tasks. However, it should be noted that the diversity of the tasks could have been further exploited by not choosing two passive visual tasks but instead an auditory task for example as the similarity of the two passive visual tasks limits the conclusions that can be drawn from the study. Also, the sample size is quite low (n=2 monkeys) and the number of trials for each task not that high (sometimes per stimulus just around 10 trials were performed). 

My main concern is that for most of the analyses it was not clear whether they have pooled data over the two monkeys. I think it is crucial to demonstrate how robust results are across the two monkeys or whether results are driven by a single monkey. Already in Fig. 2 it became clear that results are not that consistent across monkeys and I would have wished for more insights into Fig. 2C/F and 3 C/F on a single monkey level: Do both monkeys show consistency in the identified regions? For many remaining figures (Fig. 5. And Fig. 6), it was not even clear whether these were results from combined monkey data or just a single monkey selected. 

Another drawback is that most of their results center around caudal and middle 12r/46v and conclusions about the functional specialization of rostral sections of VLPFC and 45A are thus very limited. 

My other points are:

Results

- In the parcellation section of the results and methods it is mentioned that areas were also identified based on how differentially connected they are. To me it was not clear how this was achieved? Also given that their parcellation is so central to their study, I feel some more details should already be provided in the Results section about that.

- Is there a way to validate the parcellation? Given that Fig 2. And Fig. 3 already show little consistency between the monkeys, could one reason be that the anatomical parcellation actually does not translate well to a functional parcellation?

- I come from a human fMRI research background, where we expect large inter-subject variability in the functional topography of frontal regions and networks, even among healthy young adults. Is such variability not assumed in monkeys, particularly in frontal regions? How can we confidently conclude that the same anatomical region in different monkeys maps to the same functional region? Is inter-subject variability in functional organization not expected in monkeys?

- Why are different number of neurons recorded for each task?

- In the result section, it was not clear what the 2 X 12 ANOVA (Epochs, Stimuli), 3x6 ANOVA, and 9x2 ANOVA were referring to. It would help to briefly explain the different epochs defined in the main text for each task (e.g. for pictures 2 epochs: baseline and fixation and 12 different stimuli presented). In the video task description, there is also a mention of epoch 2 that the reader is not able to understand without knowing how these epochs are defined.

- Fig. 2 and 3: In the Figure legends, the reference to above and below does not seem correct (should be left and right I assume). 

- Fig. 5: To inspect the suppression better plotting a zero line would be helpful.

- Fig. 6: Is there a way to combine some of the figures from the supplementary into this figure? I feel it would already be helpful to see above threshold clusters in Fig. 6. In addition to above-chance stable coding clusters, it may also be interesting to look at dynamic coding clusters? 

- I feel the manuscript could benefit from a summary figure that relates findings to different anatomical regions.

Discussion: 

- How does the anticipation of rewards may impact the passive visual viewing task results? Would results differ if no rewards would have been given?

Methods:

- The distinction between goal-related and non-goal-related actions in videos and images was not explored. Wouldn't incorporating this dimension enhance the explanatory power of the analysis and help strengthen the conclusions of the paper?

- P 31, line 757: This should be images and not videos in the sentence.

- The input to dPCA is not well described; authors referred to an existing method but the reader should to understand the data matrix that entered the dPCA. 

Reviewer #3: Basile et al. present a detailed electrophysiological investigation into the fine-grained functional organization of the VLPFC in rhesus monkeys, leveraging a previously published cytoarchitectural and connection-based anatomical atlas. This is clearly an impressive multimodal dataset, and the effort involved in collecting electrophysiological recordings across multiple tasks from the same animals is commendable. The results are rich and generate valuable insights into the functional properties of VLPFC subregions.

However, several points would benefit from additional clarification and revision to strengthen the manuscript:

1. My main comment is that the manuscript would benefit from a clearer articulation of its main aim and how it builds on previous publications using the same dataset. The authors state that a key goal is to assess congruence between their anatomical parcellations—based on cytoarchitecture and connectivity— and functional properties. However, this does not feel like a central focus in the current version, and the manuscript lacks a strong link between the uniqueness of the parcellation and the functional findings. The introduction should better motivate why this specific parcellation is advantageous—not just by calling it "more refined," but by explaining what new insights it enables and why it's important to categorize the electrophysiological data this way. Relatedly, the manuscript would benefit from a clear hypothesis. What does the parcellation predict about the ephys results? Much of the discussion feels post-hoc and confirmatory of previous findings; if the analysis is exploratory, that should be stated explicitly. If there are a priori hypotheses, they should be clearly introduced up front.

2. The manuscript is heavily framed around a rostro-caudal organizational gradient of the LPFC. While this framework typically spans the full extent of the LPFC—from premotor areas to the frontal pole—the current results seem to reveal a much narrower gradient within the mid-VLPFC. Perhaps the authors may wish to consider alternative organizational schemes that might better account for their findings. For example, the multiple-demand framework (Crittenden & Duncan, 2012, Cerebral Cortex; Kadohisa et al, 2023 Neuron; Assem et al 2020 Cerebral Cortex) offers a contrasting view, challenging strict hierarchical models.

3. Correct me if I'm misunderstanding the task, but in the visuo-motor condition, it seems like the monkey can already decide whether to act or not during the cue period (based on the red or green cue). If that's the case, could this early decision-making explain the "static coding" observed across areas—like what's shown in the condition decodings of Figure 6? I'm just trying to reconcile this with prior work showing more dynamic coding in similar regions (e.g., Stokes et al., 2013, Neuron). Some discussion of how this task structure might influence representational dynamics would be helpful.

4. The decoding analysis looks solid overall, and the steps are well explained—but I'm a bit concerned about the statistically significant small off-diagonal clusters that show up during baseline in several areas (supp figures). Some of these are about the same size as the clusters the authors base key interpretations on. It's not clear why these baseline clusters survive if the analysis is properly controlled. Maybe it's worth double-checking the stats—do you need a stricter correction? Possibly more permutations or a higher cluster threshold to rule out spurious effects?

5. Minor Corrections and Clarifications:

* Figure 1 caption: correct "cito-architecture" to "cytoarchitecture."

* Improve figure 2 caption

* Figure 4 caption: currently missing explanation for panel C.

* Figure 5, Middle 12r plot during the object presentation phase has a mistake (avg line is in a different location than the error shades)

Reviewer #4: 

In their manuscript "Encoding of visual stimuli and behavioral goals in distinct anatomical areas of monkey ventrolateral prefrontal cortex", Basile, Gerbella and co-authors report large-scale extracellular recordings from the non-human pimate ventrolateral PFC (VLPFC) to identify a possible functional parcellation for processing visual stimuli and associated actions. Combining single-neuron and neuronal population analyses, the authors found that posterior regions strongly encoded visual stimuli, irrespective of the associated task (passive viewing or action-triggering). Internediate regions, in contrast, were strongly driven by different behavioral demands (action or "inaction"). Finally, rostral VLPFC neurons were found to be less responsive to the employed tasks. The authors interpret their results as evidence for a functional gradient in the primate PFC that unfolds along a rostro-caudal axis and matches the different anatomical connectivity profiles of the recorded regions.

This paper joins quite a large body of literature (neuroimaging and electrophysiology) on the functional parcellation of the (lateral) primate PFC, which includes previous work from the authors themselves. The state-of-the-art is nicely and scholarly summarized in the introduction. The authors highlight work by other laboratories in particular on parcellation using working memory tasks. The central motivation for the present study is to extend the investigation to more general "action organization". Indeed, the reported results largely parallel what is already known about the role of the human and non-human primate PFC in representing sensory information in a task-dependent manner and being modularly composed. In this sense, while not overly imaginative or innovative, I have no principled objections to the study's premises, the experimental approach or the conclusions the authors derive from their results. 

However, I feel the paper missed a good opportunity to provide a more encompassing, comprehensive account of the data. The results section in particular is not easy to follow—not because it presents too many surprising or diverging findings, but due to the highly granular nature of both the analyses and their descriptions. I found it quite difficult to extract the most relevant findings. A typical example of this is the population-activity section for the visuo-motor task (l. 301 ff). A more structured or streamlined (concise) presentation would improve readability and comprehension. I would like to suggest the following:

Major points:

- The current analyses could benefit from additional approaches that capture neuronal activity more comprehensively. Specifically, a geometrical analysis could help disentangle task-related variables in a more intuitive and interpretable manner (e.g. see approaches to separate sensory from choice/action information by decoding axes as in Mante et al, Nature 2013). The reliance on ANOVA alone is insufficient, as its hand-selected factors do not adequately account for the high dimensionality of the neuronal representations that are typical of the PFC.

- An unsupervised clustering analysis of neuronal response profiles should be performed to assess whether functional clusters emerge that align with known anatomical parcellation and connectivity. The current analysis groups neurons into pre-defined anatomical regions. I would like to know whether grouping by function produces similar results. Functional boundaries need not necessarily respect anatomical boundaries.

- The visuo-motor task results are described in qualitative terms without numerical quantifications. In general, no statistical tests are provided to assess whether neuronal responses are unevenly distributed across recording sites. If the authors claim a spatial organization of function within the VLPFC, they should include a test for non-uniform spatial distribution to support this conclusion. Listing (small) differences in percentages is not sufficent.

- The descriptions of experimental tasks in the main text are too rudimentary, and merely referring to the Methods section is insufficient. A brief but comprehensive summary should be provided in the main text to ensure readers can follow the rationale and design without constantly needing to refer back to the Methods.

- Likewise, several methodological aspects are not properly explained. Some examples are

"mean-net activity" (l. 303) appears to involve only baseline subtraction but does not specify whether additional normalization steps (e.g., division by variance) were applied. Clarification is needed.

"injection" is ambiguous (e.g. l. 126) - does it refer to current injection for electrolytic lesions or something else?

the type of decoding analysis (e.g., LDA, SVM etc) is not described in the Methods section

Minor points:

- Were the data sets newly acquired for this study or re-used from previous work? If existing data were analyzed, the motivation statement in the last paragraph of the Introduction should be adjusted accordingly to avoid overstating the novelty of the study

- There is a sudden shift in style starting on p. 16 (decoding results), where findings are discussed by anatomical area rather than by task or analysis, as was done in the previous sections. The authors should maintain a consistent organizational approach throughout the manuscript.

- A thorough proofread is recommended to address grammatical errors and instances of unclear or unconventional terminology, which reduce the manuscript's clarity. Some examples include:

Abstract: "pragmatic"

Introduction: "connectional evidence" 

Results: "inaction" (easily to be confused with "in action")

Caption Figure 2: referring to axes as abscissae and ordinates together with explicit naming of which variables are plotted

- The manuscript inconsistently shifts between present and past tense. The authors should ensure uniformity throughout the text, adhering to the conventional practice of using past tense for describing specific results and present tense for discussing general principles or interpretations

- The figure panels are not described in order in the captions, making it difficult to follow the visual information in a logical sequence. The descriptions should be revised to match the order in which the panels appear

- I could not follow the argument in l. 531 ff: it is my understanding that Tanila et al. and the present work both performed a posteriori matching of anatomical and functional data (?)

---

## [Editor Report · Decision Letter 2]

24 Jul 2025

Dear Dr Rozzi,

Thank you for your patience while we considered your revised manuscript "Encoding of visual stimuli and behavioral goals in distinct anatomical areas of monkey ventrolateral prefrontal cortex" for publication as a Research Article at PLOS Biology. This revised version of your manuscript has been evaluated by the PLOS Biology editors and the Academic Editor.

Based on our Academic Editor's assessment of your revision, we are likely to accept this manuscript for publication. Please also make sure to address the following data and other policy-related requests.

IMPORTANT: Please ensure that your revised manuscript incorporates the following changes.

-----------

**Financial disclosure statement:

Please add links to funding agencies in the Financial Disclosure statement in the manuscript details.

**Ethics: 

-- The Ethics statement needs to be a separate, independent (and the first) subheading in the Material & Methods section. It must include the full name of the IACUC/ethics committee that reviewed and approved the animal care and use, as well as the protocol/permit/project license number. https://journals.plos.org/plosbiology/s/ethical-publishing-practice

-- Please include the full name of the IACUC/ethics committee that reviewed and approved the animal care and use protocol/permit/project license. Please also include an approval number.

**Data:

-- Thank you for uploading your data to OSF. We have a few more requests to improve the clarity of your data availability. As currently formatted, it is not easy to tell where the relevant data from each figure is located.

---- Please cite the location of the data clearly in all relevant main and supplementary Figure legends, e.g. “The data underlying this Figure can be found in S1 Data” or “The data underlying this Figure can be found in https://doi.org/10.5281/zenodo.XXXXX and ideally, indicate where specifically in the OSF database the relevant data is found.

**Code availability:

--------------

We expect to receive your revised manuscript within two weeks. 

*Published Peer Review History*

*Press*

Sincerely,

Taylor

Taylor Hart, PhD, 

Associate Editor

thart@plos.org

PLOS Biology

---

## [Editor Report · Decision Letter 3]

28 Jul 2025

Dear Dr Rozzi,

Thank you for the submission of your revised Research Article "Encoding of visual stimuli and behavioral goals in distinct anatomical areas of monkey ventrolateral prefrontal cortex" for publication in PLOS Biology. On behalf of my colleagues and the Academic Editor, Matthew Rushworth, I am pleased to say that we can in principle accept your manuscript for publication, provided you address any remaining formatting and reporting issues. These will be detailed in an email you should receive within 2-3 business days from our colleagues in the journal operations team; no action is required from you until then. Please note that we will not be able to formally accept your manuscript and schedule it for publication until you have completed any requested changes.

PRESS

Sincerely, 

Taylor Hart, PhD, 

Associate Editor

PLOS Biology

thart@plos.org